# Toxicity of *C9orf72*-associated dipeptide repeat peptides is modified by commonly used protein tags

Javier Morón-Oset[1], Lilly KS Fischer[1], Mireia Carcolé[2,3], Ashling Giblin[2,3,4], Pingze Zhang[1], Adrian M Isaacs[2,3], Sebastian Grönke[1], Linda Partridge[1,4]

**Hexanucleotide repeat expansions in the *C9orf72* gene are the most prevalent genetic cause of amyotrophic lateral sclerosis and frontotemporal dementia. Transcripts of the expansions are translated into toxic dipeptide repeat (DPR) proteins. Most preclinical studies in cell and animal models have used protein-tagged polyDPR constructs to investigate DPR toxicity but the effects of tags on DPR toxicity have not been systematically explored. Here, we used *Drosophila* to assess the influence of protein tags on DPR toxicity. Tagging of 36 but not 100 arginine-rich DPRs with mCherry increased toxicity, whereas adding mCherry or GFP to GA100 completely abolished toxicity. FLAG tagging also reduced GA100 toxicity but less than the longer fluorescent tags. Expression of untagged but not GFP- or mCherry-tagged GA100 caused DNA damage and increased p62 levels. Fluorescent tags also affected GA100 stability and degradation. In summary, protein tags affect DPR toxicity in a tag- and DPR-dependent manner, and GA toxicity might be underestimated in studies using tagged GA proteins. Thus, including untagged DPRs as controls is important when assessing DPR toxicity in preclinical models.**

## Introduction

Amyotrophic lateral sclerosis (ALS) is a neurodegenerative disorder characterized by the progressive dysfunction and demise of specific groups of motor neurons (Taylor et al, 2016), whereas frontotemporal dementia (FTD) is a spectrum of syndromes that arises as a consequence of neurological damage in the frontal and temporal lobes (Graff-Radford & Woodruff, 2007). There are currently no disease-modifying therapies available to treat ALS and FTD, and intense research effort is devoted to better understand the underlying molecular mechanisms contributing to pathology. Expansion of a hexanucleotide repeat (HRE) GGGGCC (G4C2) in an

intronic region of chromosome 9 open reading frame 72 (*C9orf72*) is the most common genetic cause of both ALS and FTD (DeJesus-Hernandez et al, 2011; Renton et al, 2011). The number of G4C2 repeats can vary between tissues of the same affected individual, which has made it difficult to discern any associations between repeat length, phenotype, and disease severity (Beck et al, 2013; van Blitterswijk et al, 2013; Gijselinck et al, 2016; Fournier et al, 2019).

At least three independent molecular abnormalities have been associated with *C9orf72* pathogenesis in C9-ALS/FTD patients. First, haploinsufficiency of the C9orf72 protein has been detected in multiple organs and brain regions (DeJesus-Hernandez et al, 2011; Peters et al, 2015; Saberi et al, 2018), with this protein playing a role in vesicle fusion and autophagy (Nassif et al, 2017). However, C9ORF72 knock-out mice do not show signs of neurodegeneration (O'Rourke et al, 2016; Burberry et al, 2020), suggesting that haploinsufficiency is not the main disease driver. Second, C9-ALS/FTD patients display sense (G4C2) and antisense (C4G2) RNA foci (DeJesus-Hernandez et al, 2011; Lagier-Tourenne et al, 2013; Mizielinska et al, 2013). However, formation of RNA foci alone is not sufficient to cause toxicity, at least in *Drosophila* (Moens et al, 2018). Finally, five different dipeptide repeat proteins (DPR) (polyGA, polyGR, polyGP, polyPA, and polyPR) are generated by repeat-associated non-AUG-initiated (RAN) translation of the sense and antisense HRE-containing transcripts, and these accumulate in various brain regions of diseased individuals (Mori et al, 2013; Zu et al, 2013).

Postmortem analyses have revealed an apparent discrepancy between the burden of DPR aggregates, particularly abundant in the cerebellum and hippocampus, and the neurodegeneration severity, which is greatest in the cortex and spinal cord (Mackenzie et al, 2015). However, although polyGA is the most abundantly histologically detected DPR (Mori et al, 2013; Saberi et al, 2018), a recent postmortem study found that only polyGR predominantly accumulates in disease-relevant areas (Saberi et al, 2018). In contrast, pathology in mice expressing a bacterial artificial chromosome that harboured the human *C9orf72* allele with 450 G4C2 repeats was strikingly ameliorated upon treatment with GA-binding

[1]Max Planck Institute for Biology of Ageing, Cologne, Germany   [2]Department of Neurodegenerative Disease, UCL Queen Square Institute of Neurology, London, UK   [3]UK Dementia Research Institute at UCL, UCL Queen Square Institute of Neurology, London, UK   [4]Department of Genetics, Evolution and Environment, Institute of Healthy Ageing, University College London, London, UK

Correspondence: partridge@age.mpg.de

antibodies, but not a GP-specific antibody, suggesting that targeting polyGA may provide therapeutic benefits (Nguyen et al, 2019).

DPR toxicity has been extensively investigated in animal models. Most studies have shown high toxicity upon polyGR and polyPR expression (Mizielinska et al, 2014; Wen et al, 2014; Freibaum et al, 2015; Jovičič et al, 2015; Zhang et al, 2018, 2019). In contrast, no toxicity has been observed upon the expression of polyGP and polyPA (Mizielinska et al, 2014). The toxicity of polyGA is still controversial. Although there are studies that did not detect a detrimental effect of the expression of a GFP-tagged GA protein with 400 repeats (GFP-GA400) in murine neurons (Wen et al, 2014) or a GA1000-GFP in *Drosophila* survival (West et al, 2020), other studies found that untagged GA100 in flies (Mizielinska et al, 2014) and GFP-GA175 in mice (LaClair et al, 2020) decreased lifespan. In addition, brain injection of adeno-associated virus vectors encoding GFP-GA50 caused brain atrophy, and motor and cognitive deficits in mice (Zhang et al, 2016). The reason for the discrepancy in GA toxicity between these studies is currently unknown. Mechanistically, arginine-rich DPRs have been linked to, among others, translation inhibition (Zhang et al, 2018; Moens et al, 2019), impairment of nuclear cytoplasmic transport (Freibaum et al, 2015; Jovičič et al, 2015), RNA processing (Cooper-Knock et al, 2014; Conlon et al, 2016), and dysfunction of stress granule dynamics (Tao et al, 2015; Zhang et al, 2018). In contrast, GA DPRs have been shown to negatively affect proteasome activity (Zhang et al, 2016; Guo et al, 2018; Nguyen et al, 2019; Khosravi et al, 2020), DNA damage and repair (Nihei et al, 2020), nuclear cytoplasmic transport (Khosravi et al, 2017), and synaptic activity (Jensen et al, 2020).

Many studies have used DPR constructs fused at the N- or C-terminus to protein tags, including FLAG (May et al, 2014; Jovičič et al, 2015; Yang et al, 2015; Boeynaems et al, 2016), GFP (May et al, 2014; Wen et al, 2014; Xu & Xu, 2018; LaClair et al, 2020), HA (Kwon et al, 2014; Lee et al, 2017; Zhang et al, 2018; 2021), and mCherry (Lee et al, 2016; Darling et al, 2019; Morón-Oset et al, 2019; Vanneste et al, 2019). Tagging of DPRs has been useful to study their subcellular localization and aggregate morphology (May et al, 2014; Morón-Oset et al, 2019; West et al, 2020), for the identification of binding partners (May et al, 2014; Hartmann et al, 2018; Liu et al, 2022) and to elucidate subcellular localization dynamics (Jensen et al, 2020). However, whether tagged and untagged DPRs have the same molecular and pathological characteristics is not always clear.

Here, we show in vivo in the fruit fly *Drosophila melanogaster* that toxicity of arginine-rich DPRs and of polyGA is affected by the presence of the commonly used protein tags GFP, mCherry, and FLAG. Interestingly, toxicity of GR36 and PR36 proteins fused to a C-terminal mCherry tag was further increased compared with their already highly toxic untagged counterparts. In contrast, despite the toxicity of GA100 itself to lifespan, the expression of GA100-GFP and GA100-mCherry specifically in adult neurons did not reduce survival. Expression of GA100-FLAG also shortened survival, but less so than untagged GA100. Consistently, neuronal expression of untagged GA100 and GA100-FLAG led to increased expression of the stress-induced p62 protein, and exacerbated DNA damage, whereas these cellular stress responses were not induced upon expression of GA100-GFP or GA100-mCherry. The addition of the large fluorescent tags also affected the degradation propensity of GA100, whereas the smaller FLAG tag did not. In summary, our results

indicate that protein tags affect DPR toxicity in a DPR-, protein tag, and phenotype-specific manner. We suggest that experiments addressing DPR biochemistry and phenotypes should always include untagged control constructs to verify findings obtained with tagged constructs.

# Results

### An mCherry protein tag affects DPR toxicity in vivo

To address whether protein tags affect the toxicity of DPRs in vivo, we used fly lines that expressed GA, GR or PR with 36 or 100 repeats (termed hereafter GA36, GA100, GR36, GR100, PR36, and PR100), with or without a C-terminal mCherry tag. All transgenes were inserted into the attP40 docking site to minimize expression differences between the constructs (Mizielinska et al, 2014).

We initially used the GMR-Gal4 driver line to induce transgene expression specifically in the fly eye during development and early adulthood, and determined egg-to-adult survival and rough eye phenotype, read-outs frequently used to measure *C9orf72*-related toxicity (Zhang et al, 2015; Boeynaems et al, 2016). The UAS/Gal4 system is temperature-sensitive, with higher temperatures leading to increased transgene expression (Duffy, 2002). Thus, to better discern toxicity differences, flies were raised at 25°C and 29°C, corresponding to mid and high transgene expression, respectively. Eye-specific expression of either untagged or mCherry-tagged GA36 or GA100 did not affect eye size or egg-to-adult survival at 25°C or 29°C (Figs 1A–D and S1A–D). In contrast, eye-specific expression of untagged GR36 and PR36 caused a reduced eye size and affected egg-to-adult survival in a temperature-dependent manner (Figs 1A, C, and D and S1A, C, and D). Surprisingly, the expression of mCherry-tagged GR36 and PR36 led to a smaller eye size and to a further decrease in egg-to-adult survival than their untagged counterparts (Figs 1A, C, and D and S1A, C, and D). Expression of untagged GR100 and PR100 led to a very severe decrease in eye size and a strong reduction in egg-to-adult survival (Figs 1B–D and S1B–D). Eye size and egg-to-adult survival of GR100 were not further affected upon fusion with mCherry (Figs 1B–D and S1B–D). Noteworthy, while untagged and mCherry-tagged PR100 led to comparably reduced eye size and lower egg-to-adult survival at 25°C (Fig 1B–D), expression of PR100-mCherry at 29°C reduced eye size and egg-to-adult survival to a greater extent than PR100 (Fig S1B–D). In summary, no toxicity was observed upon expression of GA or GA-mCherry, whereas both GR- and PR-containing DPRs reduced egg-to-adult survival and induced disrupted eye morphology, effects that were exacerbated by tagging with mCherry.

Toxicity of DPRs can manifest differently during development and adulthood (Mizielinska et al, 2014). To address whether tagging with mCherry also affects DPR toxicity in the adult nervous system, we used the inducible elav-GeneSwitch (elav-GS) driver (Rogers et al, 2012), which upon addition of the drug RU486 induces transgene expression pan-neuronally (Figs 1E and F and S2A and B). The expression of GA36 and GA100 both reduced lifespan (Fig 1E and F). In contrast, the expression of GA36-mCherry and GA100-mCherry did not decrease lifespan (Fig 1E and F), suggesting that the addition

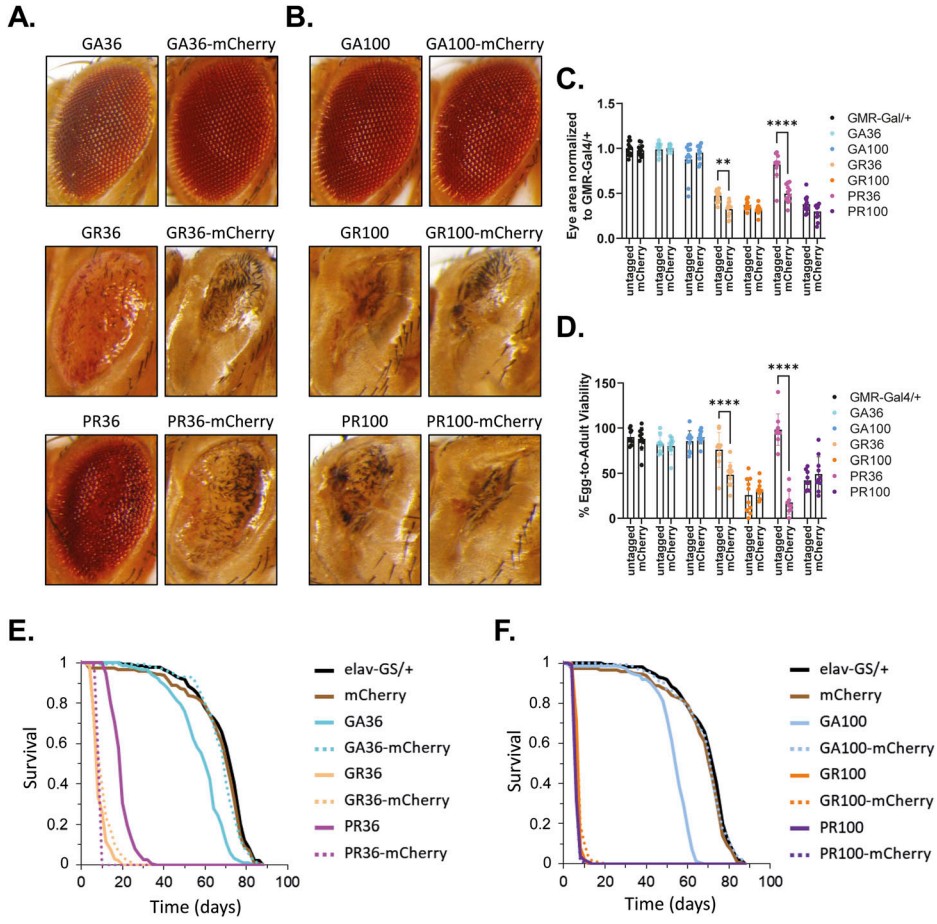

**Figure 1. Developmental and adult-onset expression of mCherry-tagged DPRs at 25°C did not trigger equal toxicity compared with untagged controls.**
**(A, B)** Representative eye images of flies expressing DPR36 or DRP36-mCherry (A) or DPR100 or DPR100-mCherry (B) at 25°C using GMR-Gal4. mCherry tagging exacerbated degenerative phenotypes of GR36 and PR36 compared with their untagged counterparts. **(A, B, C)** Quantification of eye size based on (A, B). Eye size was normalized to the eye size of GMR-Gal4/+ control flies. Tagging of GR36 and PR36 with mCherry reduced eye size compared with their untagged counterparts (two-way ANOVA + Šidák's multiple corrections test; n = 12 independent fly eyes; genotype: ****$P$ < 0.0001; mCherry tagging: ****$P$ < 0.0001; interaction: ****$P$ < 0.0001). **(D)** Egg-to-adult viability of flies expressing GA, GR, and PR with 36 or 100 repeats, untagged or tagged with mCherry. Egg-to-adult survival of flies expressing mCherry-tagged GR36 and PR36 was significantly reduced compared with their untagged counterparts (two-way ANOVA + Šidák's multiple corrections test; n = 9–10 independent vials; genotype: ****$P$ < 0.0001; mCherry tagging: ****$P$ < 0.0001; interaction: ****$P$ < 0.0001). **(E)** Survival curves of RU486-fed flies induced to express the pan-neuronal elav-GS driver in combination with no transgene (elav-GS/+), mCherry, DPR36 or DPR36-mCherry transgenes. Pairwise comparisons with multiple-testing correction showed that mCherry expression did not affect fly lifespan (elav-GS/+ versus mCherry; log-rank + Bonferroni's multiple corrections test; $P$ > 0.05), GA36 was more toxic than GA36-mCherry (GA36 versus GA36-mCherry; log-rank + Bonferroni´s multiple corrections test; ****$P$ < 0.0001), GR36 was highly toxic and equally toxic to GR36-mCherry (GR36 versus GR36-mCherry; log-rank + Bonferroni's

multiple corrections test; $P$ > 0.05) and PR36 was also highly toxic with its toxicity reduced compared with PR36-mCherry (PR36 versus PR36-mCherry; log-rank + Bonferroni's multiple corrections test; ****$P$ < 0.0001). **(F)** Survival curves of RU486-fed flies induced to express the pan-neuronal elav-GS driver in combination with no transgene (elav-GS/+), mCherry, DPR100 or DPR100-mCherry transgenes. Pairwise comparisons with multiple-testing correction revealed that GA100 was more toxic than GA100-mCherry (GA100 versus GA100-mCherry; log-rank + Bonferroni's multiple corrections test; ****$P$ < 0.0001), whereas GR100 and PR100 were equally toxic to GR100-mCherry and PR100-mCherry, respectively (GR100 versus GR100-mCherry and PR100 versus PR100-mCherry; log-rank + Bonferroni's multiple corrections test; $P$ > 0.05). The uninduced EtOH controls of all genotypes are shown in Fig S2.

of the C-terminal mCherry-tag blocks the toxicity of GA-containing DPRs. Consistent with previous findings (Mizielinska et al, 2014), adult-onset, neuronal expression of GR36 and GR100 strongly reduced survival of female flies (Fig 1E and F), and the presence of an mCherry tag did not further affect survival (Fig 1E and F). PR36 expression also shortened lifespan (Fig 1E), and flies expressing PR36-mCherry were even shorter lived (Fig 1E). PR100 and PR100-mCherry induced similarly shortened lifespans (Fig 1F). These data indicate that the addition of an mCherry tag affects in vivo toxicity of DPRs in a DPR-dependent manner, increasing the toxicity of arginine-containing DPRs and reducing the toxicity of GA-containing DPRs.

We next investigated whether the increased toxicity of mCherry-tagged GR and PR might be caused by differences in DPR protein level or subcellular localization. Therefore, we performed immunostainings on fly brains after 3 d of pan-neuronal, transgene induction. GR signal was strongly enriched in the median neurosecretory cell (MNC) region (Fig S3A and C), consistent with our previous findings (Morón-Oset et al, 2019). Both the GR36 and

GR100 signals were exclusively nuclear, and there were no obvious morphological differences between untagged and mCherry-tagged polyGR (Figs 2A and S3E). Quantification of GR protein expression in the pars intercerebralis region showed no differences in protein levels between untagged and mCherry-tagged polyGR (Fig 2C). We further assessed total GR levels in heads of flies after 3 d of pan-neuronal, transgene expression using a quantitative Meso Scale Discovery (MSD) immunoassay (Moens et al, 2018; Atilano et al, 2021). In agreement with our immunostaining results, we found no effect of mCherry-tagging on GR protein levels (Fig 2D). Thus, mCherry-tagging did not affect the subcellular localization or protein levels of polyGR, suggesting that the increased toxicity is not caused by more toxic protein present in cells.

PolyPR was detected throughout the fly brain and also accumulated in the MNC region, on which we focused for further analysis (Fig S3B and D). The PR36 signal was exclusively nuclear and diffuse, whereas PR100 showed a punctated pattern and accumulated both inside and outside of the nucleus (Figs 2B and

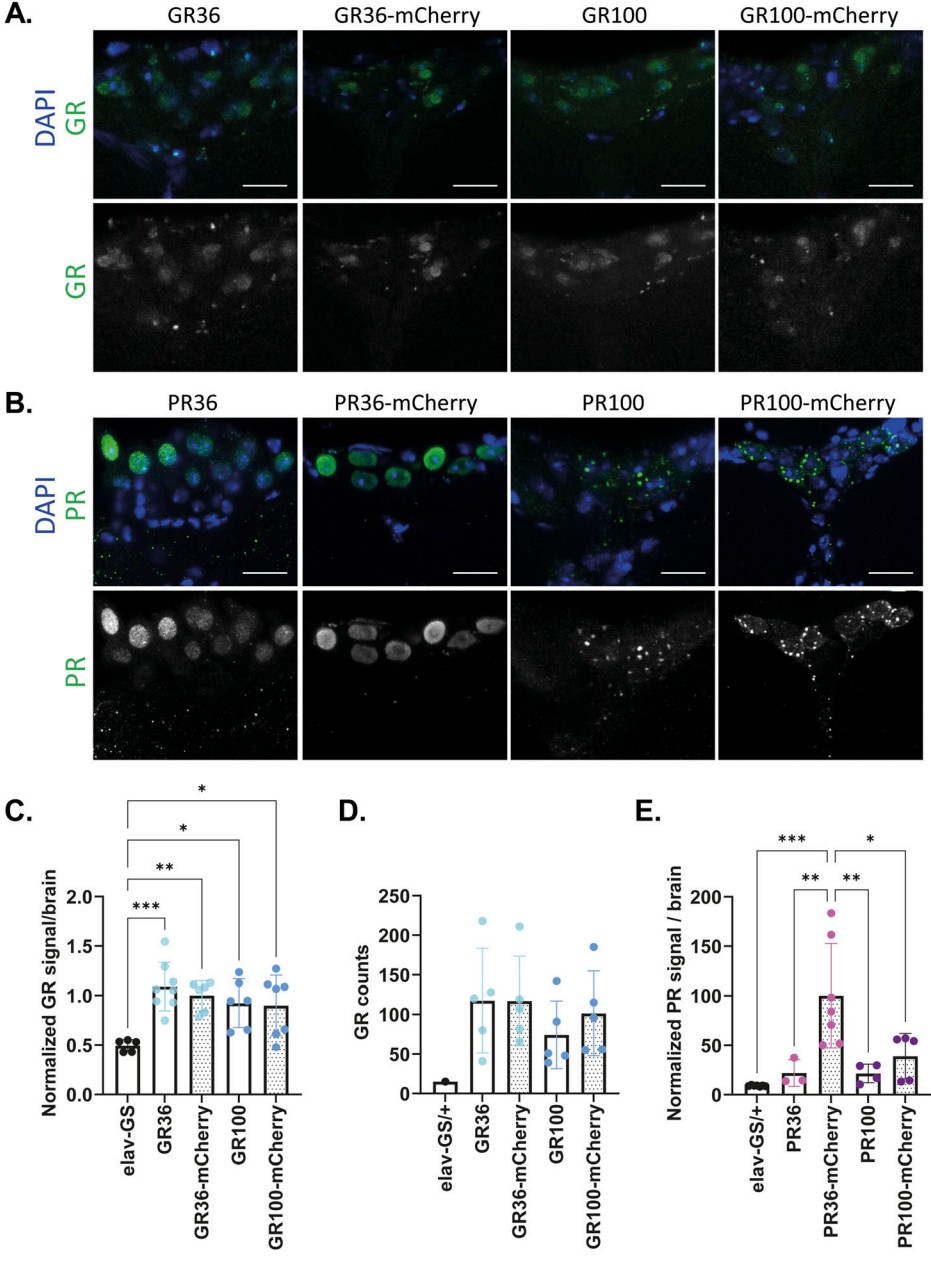

**Figure 2. Tagging of polyGR or polyPR to mCherry did not affect subcellular localization or morphology, but it increased PR protein expression in PR36 flies.**

**(A, B)** Representative, high-magnification images of the pars intercerebrallis region from the brains of untagged and mCherry-tagged polyGR (A) or polyPR (B) treated with RU486 to express the indicated constructs under the pan-neuronal driver elav-GS for 3 d. Both untagged and mCherry-tagged polyGR and PR36 were intranuclear and diffused. However, PR100 and PR100-mCherry were both nuclear and cytoplasmic. GR and PR morphologies were not affected by mCherry-tagging. **(A)** An anti-GR antibody and equal imaging settings were used across genotypes. **(B)** An anti-PR antibody was used. Equal settings were used in PR36-mCherry, PR100, and PR100-mCherry flies, whereas higher settings were used in PR36 and elav-GS/+ to increase PR detectability (B). Scale bars in images are 10 μm. Elav-GS/+ controls are shown in Fig S3. **(C)** Quantification of GR intensity in pars intercerebrallis region showed that mCherry-tagging did not affect GR levels (one-way ANOVA + Tukey's multiple comparisons test; n = 4–6 brains; $**P < 0.01$). **(D)** GR dipeptide levels in heads from untagged and tagged polyGR flies determined by MSD immunoassay were unchanged upon mCherry-tagging after 3 d of transgene induction (one-way ANOVA + Tukey's multiple comparisons test; n = 5 sets of 12–20 fly heads; $P > 0.05$). Values were blank-corrected. **(E)** Quantification of PR intensity in pars intercerebrallis region showed that mCherry-tagging particularly increased PR levels of PR36 (one-way ANOVA + Tukey's multiple comparisons test; n = 3-7 brains; $***P < 0.001$). PR was difficult to detect in most PR36 flies. Equal settings were used for PR signal quantification.

S3F). Although protein length affected the subcellular localization of PR, mCherry-tagging did not overtly affect the morphology of PR36 or PR100 (Fig 2B). However, the brains of PR36-mCherry–expressing flies showed higher PR levels than those of untagged PR36 flies (Fig 2E). In agreement, we found a non-significant trend towards increased PR protein levels in PR100-mCherry flies compared with PR100 flies (Fig 2E). We performed MSDs using anti-PR antibodies, but we were unable to reliably measure PR levels using this technique (Fig S3G). Overall, our results suggest that mCherry-tagging may lower the turnover of polyPR, which might contribute to its increased toxicity.

## GA100 toxicity is reduced by protein tags in a tag-specific manner

We next addressed whether DPR toxicity was dependent on the type of protein tag fused to the DPR. Given the obvious difference in survival between mCherry-tagged and untagged GA100, we focused on GA100 to address this question and used lifespan as read-out. In addition to the previously used mCherry tag (Morón-Oset et al, 2019), we generated flies expressing GA100 fused to GFP (GA100-GFP) and FLAG (GA100-FLAG), two commonly used protein tags to address DPR toxicity (Wen et al, 2014; Jovičič et al, 2015; Boeynaems et al, 2016; West et al, 2020). All transgenes were inserted into the

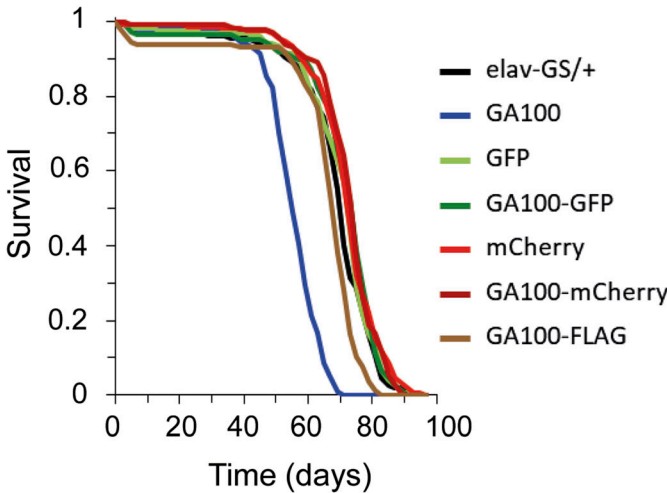

**Figure 3. Neuronal expression of tagged GA100 does not decrease fly survival to the same extent as untagged GA100.**
Survival curves of flies treated with RU486 to pan-neuronally express each of the indicated genotypes with elav-GS. Pairwise comparisons with multiple-testing correction showed that GA100 shortened fly lifespan compared with driver-only or all tagged GA100 constructs (GA100 versus elav-GS/+, GA100 versus GA100-GFP, GA100 versus GA100-mCherry, GA100 versus GA100-FLAG; log-rank + Bonferroni's multiple corrections test; ****$P < 0.0001$), and this toxicity was completely rescued by tagging with GFP (elav-GS/+ versus GA100-GFP; log-rank + Bonferroni's multiple corrections test; $P > 0.05$) or mCherry (elav-GS/+ versus GA100-mCherry; log-rank + Bonferroni's multiple corrections test; $P > 0.05$). GA100-FLAG also induced a reduction in lifespan compared with driver-only flies (GA100-FLAG versus elav-GS/+; log-rank + Bonferroni's multiple corrections test; ***$P < 0.001$), but less so than GA100 (GA100 versus GA100-FLAG; log-rank + Bonferroni's multiple corrections test; ****$P < 0.0001$). The uninduced EtOH controls of all genotypes are shown in Fig S4.

attP2 docking site to minimize expression differences between the lines (Figs 3 and S4A and B). Adult-onset, neuron-specific expression of untagged GA100 by elav-GS shortened fly survival compared with control flies, consistent with previous results (Mizielinska et al, 2014). In contrast, expression of GA100-mCherry did not affect survival (Fig 3). Flies expressing GA100-GFP were longer lived than those expressing GA100, and showed similar survival to flies expressing GA100-mCherry or control flies (Fig 3), demonstrating that, as with mCherry, GFP tagging interfered with the toxicity of GA100. In contrast, flies expressing GA100-FLAG were significantly shorter lived than controls, but not as short lived as flies expressing untagged GA100 (Fig 3).

Tagging with GFP and mCherry thus abolished GA100-mediated lifespan shortening, whereas FLAG-tagging lessened the detrimental effects of expression of GA100. These results show that GA toxicity is strongly influenced by commonly used tags in vivo.

## Degradation of GA100 is affected by protein tags

We next investigated whether tags affected the subcellular localization, protein levels, solubility or degradation propensity of DPRs, because such effects could account for the modification of toxicity by tags. First, we performed brain immunostainings of flies induced to express untagged or tagged GA100 for 1 d. We detected punctated GA aggregates in the brains of GA100 and GA100-FLAG flies (Fig S5A),

in line with previous reports (Morón-Oset et al, 2019; Jensen et al, 2020; Nihei et al, 2020). We detected no nuclear GA deposits in the large MNCs of the brains of GA100 and GA100-FLAG flies (Fig S5B), indicating that GA exclusively localized to the cytoplasm. In addition, although the brains of GA100-GFP and GA100-mCherry flies showed GFP- and mCherry-positive aggregates, they also showed diffused GA signal that did not co-localize with the naked GFP or mCherry fluorescent signals, respectively (Fig S5A and B), suggesting that their diffuse GA signals partially consisted of degradation products. The high density of GFP- and mCherry-positive inclusions in the brains of GA100-GFP and GA100-mCherry flies may explain the limited penetration of our anti-GA antibody (Fig S5B), which had previously been observed in cells using a different anti-GA antibody (Liu et al, 2022). Importantly, no nuclear GA or fluorescent tag-specific signals were detected in the brains of GA100-GFP or GA100-mCherry flies (Fig S5B), suggesting that the subcellular localization of these constructs is exclusively cytoplasmic in fly brains regardless of the presence or absence of protein tags, as previously shown (Mizielinska et al, 2017; Morón-Oset et al, 2019).

We next compared protein levels of untagged and tagged GA100 by Western blot analysis of protein extracts isolated from heads of flies expressing GA100, GA100-GFP, GA100-mCherry, and GA100-FLAG, and fed for 1, 5 or 25 d with the inducing agent RU486. Using an anti-GA antibody, discrete, low molecular weight (LMW) bands were detected for all constructs after short-term induction for 1 d, with GA100-GFP and GA100-mCherry showing higher protein abundance than untagged GA100 and GA100-FLAG (Fig S6A). We also detected full-length, LMW FLAG-, GFP-, and mCherry-tagged GA100 using tag-specific antibodies (Fig S6A). Unlike in GFP- and mCherry-only extracts, anti-GFP and anti-mCherry antibodies also revealed high molecular weight (HMW) in GA100-GFP and GA100-mCherry extracts, respectively (Fig S6A), suggesting that GA100-GFP and GA100-mCherry are aggregation-prone, in line with previous reports (Zhang et al, 2016; 2021). Longer exposure of the Western blot membrane probed with anti-GA revealed a smear downwards of the GA100-GFP and GA100-mCherry bands, which was not observed for either GA100 or GA100-FLAG (Fig S6A). In line with our immunostaining results, the smear suggests increased degradation of the GA100-GFP and GA100-mCherry proteins. By days 5 and 25, GA100-GFP and GA100-mCherry extracts showed HMW bands, and LMW full-length and degradation bands detected by anti-GA in GA100-GFP and GA100-mCherry after 5 or 25 d of transgene induction (Fig S6B). GFP- and mCherry-specific bands were also observed in extracts from GA100-GFP and GA100-mCherry flies (Fig S6B), suggested partial cleavage after 5 and 25 d of transgene expression as previously observed (Zhang et al, 2021). In contrast, no degradation bands were observed in GA100 or GA100-FLAG extracts even after prolonged expression for 25 d, by which only HMW bands were detected (Fig S6B). These Western blot results suggest that, unlike FLAG-tagging, fusion of fluorescent tags to GA100 increases its solubility and degradation propensity. However, the difference in solubility between untagged and mCherry-tagged GA100 could not be verified by MSD immunoassay, which showed no significant difference between the soluble and insoluble fractions of GA100 and the tagged GA100 constructs (Fig S6C and D). Thus, it is currently unclear whether the fluorescent tags affect the solubility of GA100. However, in none of the assays we detected significant lower levels of the tagged GA100 protein, indicating that the lack of toxicity

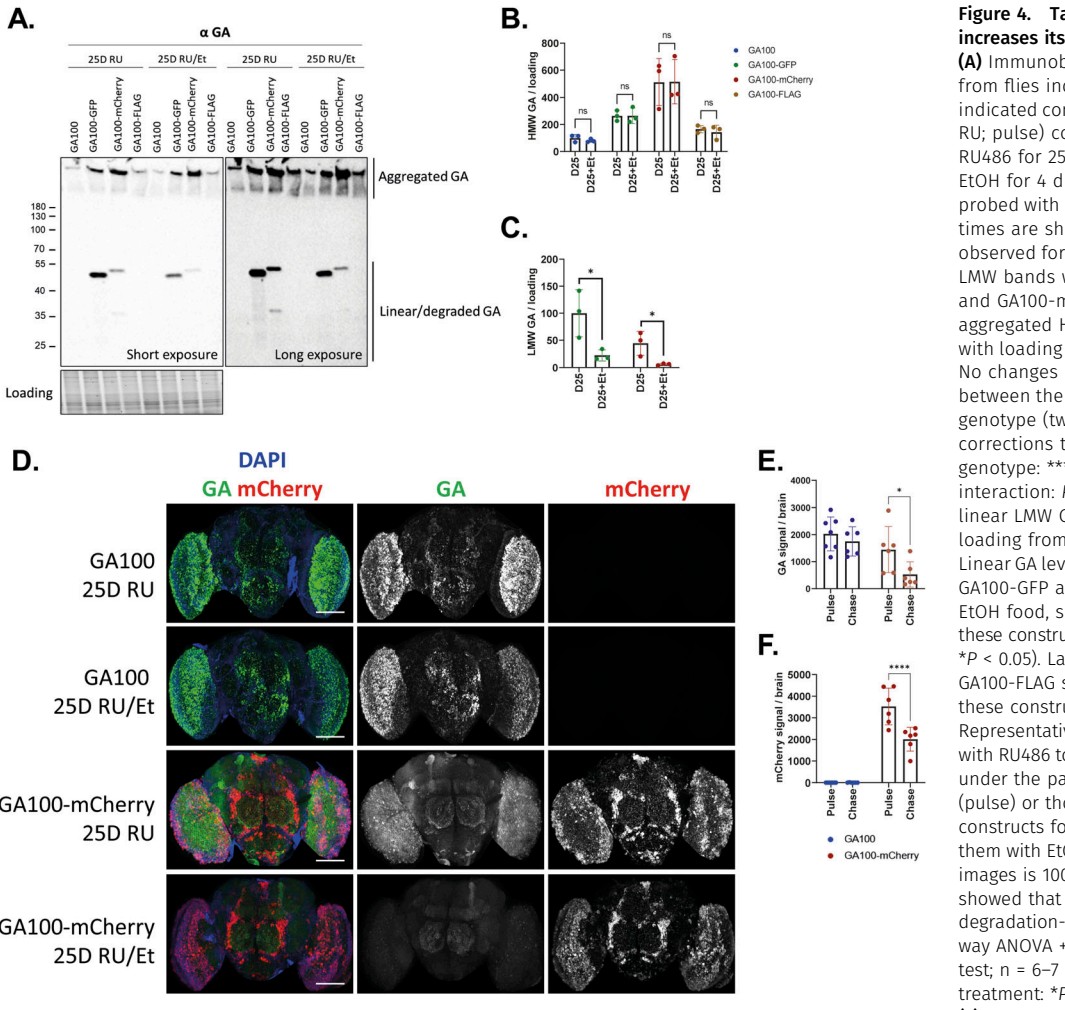

**Figure 4. Tagging of GA100 to fluorescent tags increases its degradation propensity.**
**(A)** Immunoblotting of head protein extracts from flies induced to express each of the indicated constructs with elav-GS for 25 d (25D RU; pulse) compared with flies treated with RU486 for 25 d and subsequently transferred to EtOH for 4 d (25 RU/Et; chase). Samples were probed with anti-GA and two different exposure times are shown. Aggregated bands were observed for each of the indicated genotypes. LMW bands were only observed for GA100-GFP and GA100-mCherry. **(A, B)** Quantification of aggregated HMW GA protein levels compared with loading from immunoblotting shown in (A). No changes in aggregated GA were detected between the pulse and the chase of the same genotype (two-way ANOVA + Šidák's multiple corrections test; n = 3 sets of 20 fly heads; genotype: ****$P$ < 0.0001; treatment: $P$ > 0.05; interaction: $P$ > 0.05). **(A, C)** Quantification of linear LMW GA protein levels compared with loading from immunoblotting shown in (A). Linear GA levels were reduced in flies expressing GA100-GFP and GA100-mCherry after 4 d on EtOH food, suggesting early degradation of these constructs (t test; n = 3 sets of 20 fly heads; *$P$ < 0.05). Lack of soluble bands for GA100 and GA100-FLAG suggests impaired degradation of these constructs after 4 d of chase. **(D)** Representative images of adult fly brains treated with RU486 to express the indicated constructs under the pan-neuronal driver elav-GS for 25 d (pulse) or those expressing the indicated constructs for 25 d and subsequently treating them with EtOH food for 4 d (chase). Scale bar in images is 100 $\mu$m. **(E)** Quantification of GA signal showed that GA100-mCherry is more degradation-prone than untagged GA100 (two-way ANOVA + Bonferroni's multiple corrections test; n = 6–7 brains; genotype: **$P$ < 0.01; treatment: *$P$ < 0.05; interaction: $P$ > 0.05). **(F)** Quantification of mCherry signal showed that GA100-mCherry is degradation-prone (Two-Way ANOVA + Bonferroni's multiple corrections test; n = 6-7 brains; genotype: ****$P$ < 0.0001; treatment: ***$P$ < 0.001; interaction: ***$P$ < 0.001).

upon expression of GA100 fused to large fluorescent tags may not be attributable to reduced protein levels.

Because we detected evidence for increased degradation in GA100-GFP and GA100-mCherry flies by immunostainings and Western blot, we next performed a pulse-and-chase experiment to address this further. Flies were fed with RU486 for 25 d to induce transgene expression (pulse). Subsequently, flies were transferred to EtOH control food for 4 d to stop transgene expression (chase). LMW and HMW protein levels were compared using Western blot and a GA-specific antibody. The intensity of the HMW bands did not change between the pulse and the chase for any of the constructs (Fig 4A and B), suggesting that the aggregated forms of both un-tagged and tagged GA100 are stable and not degraded within the 4-d observation period. In contrast, we observed reduced levels of the LMW form of GA100-GFP and GA100-mCherry between the chase and the pulse (Fig 4A and C), suggesting reduced stability of the linear isoforms of GA100-GFP and GA100-mCherry. In contrast, we did not detect LMW forms of GA100 and GA100-FLAG at this age and all detected GA for these genotypes were similarly insoluble during the pulse and the chase (Fig 4A and B), suggesting that all remaining

GA in GA100 and GA100-FLAG flies after 25 d of expression are resistant to degradation for at least 4 d. To further confirm these results, we performed a pulse-and-chase experiment with GA100 and GA100-mCherry and used immunostainings on fly brains as read-out. In agreement with our Western blot results, the amount of GA signal in GA100 fly brains was unchanged after the 4-d chase (Fig 4D and E). In contrast, the GA (Fig 4D and E) and mCherry (Fig 4D and F) signals were significantly reduced during the chase in brains expressing GA100-mCherry, suggesting that GA100-mCherry is less stable than untagged GA100. In summary, our data suggest that GA100 tagged with GFP or mCherry is less stable than GA100, which might contribute to the lower toxicity of fluorescently la-belled GA. In contrast, the smaller FLAG tag had no effect on the protein stability of GA100.

### The proteasome contributes to the degradation of GA100-mCherry

To investigate whether the proteasome or autophagy plays a role in the increased degradation propensity of fluorescent tag-fused

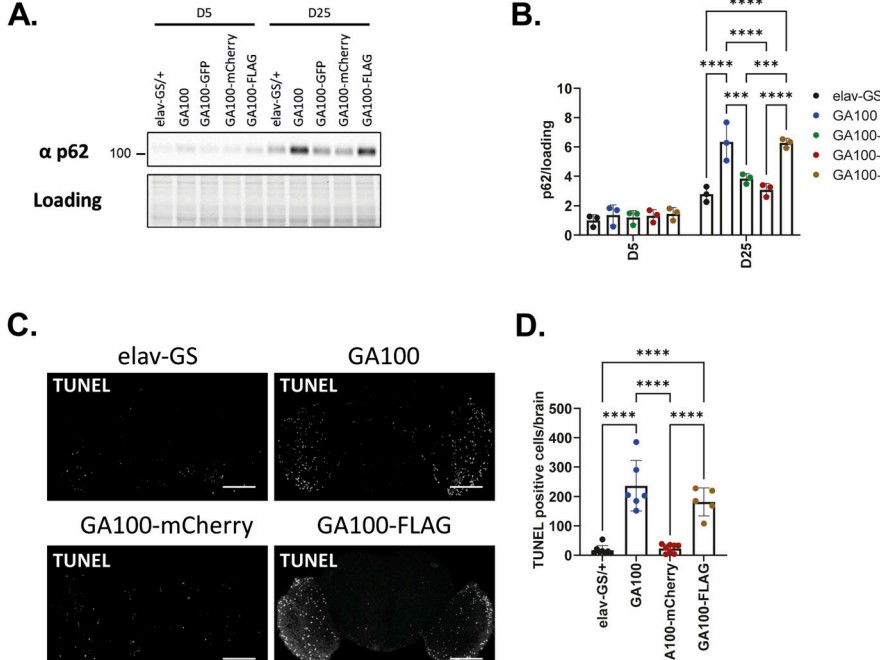

**Figure 5. P62 protein levels and DNA damage are similarly increased in GA100 and GA100-FLAG but not upon GA100-GFP and GA100-mCherry induction.**
**(A)** Immunoblotting of head protein extracts from flies induced to express each of the indicated constructs under elav-GS for 5 or 25 d. Extracts were probed with an anti-p62 antibody. **(A, B)** Quantification of p62 protein levels compared with loading from immunoblotting shown in (A). No protein changes across genotypes were observed at 5 d of induction, but p62 levels were increased in GA100 and GA100-FLAG extracts compared with driver alone, GA100-GFP and GA100-mCherry at 25 d of induction (two-way ANOVA + Tukey's multiple corrections test; n = 3 sets of 20 fly heads; genotype: ****$P < 0.0001$; age: ****$P < 0.0001$; interaction: ****$P < 0.0001$). **(C)** Representative images of adult fly brains upon expression of the indicated constructs under elav-GS for 25 d. Brains were stained with the TUNEL dye (green) and subsequently imaged. Scale bar in images is 100 $\mu m$. **(D)** Quantification of the number of TUNEL positive cells per brain (One-way ANOVA + Tukey's multiple comparisons test; n = 5–8 brains; ****$P < 0.0001$).

GA100 constructs, we measured GA100-mCherry levels in flies with reduced activity of the proteasome or autophagy. Therefore, GA100-mCherry was co-expressed with RNAi against the autophagy regulator atg5 (atg5 RNAi), which blocks the autophagic flux (Bjedov et al, 2020; Lu et al, 2021; Juricic et al, 2022), or RNAi against the proteasome subunit α5 (Pros α5 RNAi). Pros α5 is a proteasome subunit of the 20S catalytic core required for the hydrolytic activity of the 26S proteasome machinery, which is key for tissue integrity in *Drosophila* (Velentzas et al, 2013). Pros α5 RNAi for 5 d reduced Pros α5 transcript levels (Fig S7A) and strongly increased the protein levels of p62 (Fig S7B and C), a stress-induced, autophagy-associated protein (Lamark & Johansen, 2012) whose up-regulation is a common response to proteasome inhibition to enable the autophagic clearance of aggregated proteins (Nezis et al, 2008; Lőw et al, 2013; Velentzas et al, 2013), thus validating the functionality of Pros α5 RNAi. Neuron-specific co-expression of GA100-mCherry and atg5 RNAi for 15 d only had a very mild and nonsignificant effect on the GA or mCherry signals in fly brains (Fig S8A–C). In contrast, there was a significant increase in GA and mCherry signals in the brains of flies co-expressing GA100-mCherry and Pros α5 RNAi (Fig S8A–C), indicating that GA100-mCherry is more stable when proteasome activity is inhibited.

In summary, these data suggest the proteasome as the primary proteolytic system contributing to the degradation of GA100-mCherry, which may contribute to the lower toxicity of the fluorescently labelled GA100 proteins.

### Neuronal expression of GA100 induces cellular stress that is lessened by large protein tags

We next addressed whether the molecular responses of brain cells towards GA expression is altered by the protein tags. In ALS/FTD patient tissue, DPRs co-localize with p62 (Schludi et al, 2015), and in mammals only GA, but not the other DPRs, increase expression of p62 (May et al, 2014; Zhang et al, 2016), indicating that p62 is a relevant cellular response after GA expression. The fly homologue of p62 is refractory to sigma P (Ref(2)P), and its level has been shown to increase during normal fly ageing (Nezis et al, 2008; Bartlett et al, 2011) and to reduce toxicity after age-dependent stress (Aparicio et al, 2019). Consistently, we observed an age-related increase in p62 levels between 5 and 25 d for all tested genotypes using a p62-specific antibody in Western blot analysis of fly heads (Fig 5A and B). While the expression of untagged GA100 did not affect p62 protein levels in young flies after 5 d of induction, p62 protein levels were significantly increased compared with control flies upon induction of GA100 for 25 d (Fig 5A and B). Thus, as in mammals, GA expression induces p62 in *Drosophila*. Expression of GA100-FLAG also increased p62 protein levels to a similar extent as does untagged GA100 (Fig 5A and B). In contrast, expression of mCherry- or GFP-tagged GA100 for 25 d did not result in increased p62 levels (Fig 5A and B), consistent with the lack of toxicity observed upon expression of these constructs. Thus, the large, fluorescent protein tags interfered with the induction of p62 by GA100, whereas the smaller FLAG tag did not.

Another common effect of GA expression in mammals is the induction of DNA damage (Nihei et al, 2020). To assess whether the tags modulated GA-mediated DNA damage, we used a TUNEL assay on fly brains after 25 d of transgene induction to quantify the number of TUNEL-positive cells per brain. Because the TUNEL excitation and emission spectra overlap with those of GFP, we focused on untagged GA100, GA100-mCherry, and GA100-FLAG for this experiment. Expression of untagged GA100 significantly increased the number of TUNEL-positive cells (Fig 5C and D), suggesting that GA expression induced DNA damage in fly brains. GA-induced DNA

fragmentation was mainly observed in the optic lobes. Consistently, expression of the poly-glutamine-containing ataxin 7 protein (Latouche et al, 2007) also caused DNA fragmentation preferentially in the optic lobes, indicating that this brain area is specifically prone to accumulate DNA damage upon expression of aggregation-prone pathological proteins. The expression of GA100-FLAG also significantly increased the number of TUNEL-positive cells to a similar degree as the expression of untagged GA100 (Fig 5C and D). In contrast, GA100-mCherry expression did not increase the number of TUNEL-positive cells compared with control flies (Fig 5C and D).

Thus, stress-associated molecular changes induced by GA100 expression were prevented by adding large fluorescent tags but not by the smaller FLAG tag, consistent with the finding that GA100-mCherry or GA100-GFP were not toxic to lifespan, unlike their untagged counterparts. However, the finding that expression of GA100-FLAG induces the same molecular changes as untagged GA100, but is not as toxic to lifespan, suggests that molecular changes other than DNA damage or p62 induction underlie the lifespan difference.

## Discussion

Gaining a thorough understanding of the molecular underpinnings of DPR toxicity is relevant to develop new therapeutic strategies for *C9orf72* mutation carriers. To this end, preclinical animal models expressing DPRs are often used, with protein tagging for better detection. However, whether these protein tags affect DPR toxicity has not been systematically addressed. In the current study, we used the fruit fly *Drosophila* to study whether commonly used protein tags affect the toxicity of the *C9orf72*-associated DPRs in vivo. We show that a C-terminal mCherry tag increased in vivo toxicity of arginine-rich DPRs, whereas it reduced toxicity of polyGA. The large, fluorescent tags GFP and mCherry affected polyGA toxicity more than the smaller FLAG tag, demonstrating that protein tags affect DPR toxicity in a tag- and DPR-specific manner. Consistently, while untagged and FLAG-tagged polyGA increased p62 levels and DNA damage, this was not observed in polyGA tagged with the large fluorescent tags. Thus, both phenotypic read-outs and cellular responses after DPR expression are affected by protein tags, suggesting that preclinical studies should always include untagged DPR controls.

Most studies have reported high levels of toxicity after the expression of both untagged and tagged polyGR and polyPR (Mizielinska et al, 2014; Wen et al, 2014; Freibaum et al, 2015; Jovičič et al, 2015; Boeynaems et al, 2016; Lopez-Gonzalez et al, 2016). Our findings show that a C-terminal mCherry-tag slightly increased toxicity of GR36 and polyPR, especially upon high levels of eye-specific expression of these constructs. The higher toxicity of polyPR-mCherry might be due to an increase in protein levels. In contrast, the higher toxicity of GR36-mCherry cannot be explained by increased protein levels, as there was no difference observed either by immunohistochemistry or by MSD immunoassay. Thus, fluorescent tags can have different effects on DPR stability dependent on the DPR to which they are attached. Although it is currently unclear what causes the increase in toxicity

upon expression of GR36-mCherry, mCherry-tagging may alter the interaction of GR with its binding partners and thereby cause more severe effects. Noteworthy, no increase in toxicity was observed when GR100 was tagged with mCherry, which might be due to the already very high baseline toxicity of this DPR. Interestingly, the expression of GR36 and PR36 fused to an N-terminal GFP-tag shortened survival of flies to a lesser extent than the expression of their untagged counterparts (Xu & Xu, 2018), indicating reduced toxicity. This finding suggests that the position of the tag can affect toxicity of arginine-rich DPRs differentially depending on whether the tag is located at the N- or C-terminus.

Whether and to what extent expression of polyGA is toxic is still disputed. Expression of GA175-GFP in immortalized cells (May et al, 2014), GA50-FLAG in yeast (Jovičič et al, 2015; Boeynaems et al, 2016), GA1000-GFP in *Drosophila* (West et al, 2020) and GFP-polyGA in primary cortical and motor neurons (Wen et al, 2014) did not affect cell viability or fly survival, which is in agreement with our current fly study using GA100-GFP and GA100-mCherry. In contrast, transfection of immortalized cells with GA50-V5 and GA50-HA (Zhang et al, 2014), pan-neuronal expression of untagged GA100 (Mizielinska et al, 2014) and GA100-FLAG in *Drosophila*, expression of GA149-myc and GA149-GFP in primary rat neurons (May et al, 2014), adeno-associated virus-mediated expression of GFP-GA50 in mouse brain (Zhang et al, 2016) and pan-neuronal, constitutive expression of GA175-CFP using the Thy1 promoter (Schludi et al, 2017) caused toxicity. Surprisingly, pan-neuronal constitutive expression of GFP-GA175 in mice using the CAG promoter led to high levels of toxicity, which even exceeded that of GFP-PR175 in congenic mice (LaClair et al, 2020). Differences in the experimental systems, repeat length, transgene expression levels, and nature or absence of protein tags may be important factors determining polyGA toxicity. Here, we simultaneously compared the toxicity of tagged and untagged polyGA in *Drosophila* and showed that C-terminal protein tags reduce the toxicity. Thus, including an untagged GA control is essential for studies that address GA toxicity.

We observed major protein degradation and lower protein stability of the tandem GFP- or mCherry-linked GA100 constructs compared with untagged GA100, which may contribute to their lack of toxicity. Previous studies used anti-GFP to detect their GFP-containing polyGA constructs (Zhang et al, 2016; LaClair et al, 2020), thus precluding the assessment of whether major GA degradation also occurs in independent constructs. We found that lowering the activity of the proteasome via Prosα5 RNAi increased the levels of GA and mCherry signals upon GA100-mCherry co-expression, implicating the proteasome in the degradation of fluorescently labeled GA100. Interestingly, GFP-GA175 forms polyGA ribbons surrounded by proteasomes in neurons, leading to an increased proportion of proteasomes in both quiescent ground and substrate-processing states. This indicates that GFP-GA175 triggers proteasomal inhibition (Guo et al, 2018). Indeed, GFP-tagged polyGA increased accumulation of proteasome substrates in cultured cells (May et al, 2014; Zhang et al, 2016; Khosravi et al, 2020). However, to the best of our knowledge, no study has verified these effects with untagged constructs, which would be relevant for therapeutic testing. Our data are in agreement with untagged GA100 reducing proteasomal activity, as p62 protein levels accumulated faster upon

GA100 and GA100-FLAG expression, and higher p62 levels correlate with reduced proteasomal activity (Velentzas et al, 2013). Alternatively, degradation-independent conformational changes of GA100 after fusion to large tags may underpin changes in toxicity. For instance, N-terminal linkage of β-actin to GFP led to excessive formation of microfilaments (Nagasaki et al, 2017), and GFP-tagging of the microtubule-associated protein τ dramatically lowered its aggregation propensity (Kaniyappan et al, 2020).

Comparative quantification of GA levels in extracts from flies expressing untagged and tagged GA100 was not trivial, as anti-GA failed to accurately detect GA aggregates in the brains of GA100-GFP and GA100-mCherry flies using brain immunostainings. In addition, although our Western blot analysis showed markedly higher GA levels in head extracts from GFP- and mCherry-tagged GA100 flies, particularly after the quantification of GA LMW bands, our MSD analysis indicated that tags did not alter soluble or insoluble GA levels. Given that we used a slow centrifugation speed to prepare protein extracts for Western blotting, we hypothesize that a proportion of the most aggregated GA may have been pelleted and, therefore, not efficiently isolated. Furthermore, given that our MSD experiments indicated that the soluble, GA levels in untagged GA100 extracts did not differ from those of tagged GA100 extracts, the LMW GA bands observed by Western blot may consist of soluble and partially insoluble proteins. However, because we only used two fractions (i.e., soluble and insoluble), we cannot rule out whether tags (especially, but not only, the fluorescent tags) may partially affect the solubility of GA100, for which more than two protein fractions would need to be assessed. Of note, even though the same antibody was used among all three techniques, we cannot rule out whether the efficiency of our ECL-labelled (SULFO-TAG) detection antibody to elicit light and, therefore, detect GA, may be partially affected by protein tags.

GA100-FLAG was less toxic than untagged GA100 but more toxic than GA100-GFP and GA100-mCherry. Ubiquitous FLAG-GA50 expression was also reported to exert mild toxicity in female flies (Boeynaems et al, 2016). Along with increased p62 protein levels, the expression of GA100 and GA100-FLAG also caused comparable accumulation of DNA damage in fly brains, which we did not detect in flies expressing GA100-mCherry. The ability of polyGA to cause DNA damage has been controversial, with short-term expression of HA-GA80 not triggering DNA damage in control-induced pluripotent stem cells-derived motor neurons (Lopez-Gonzalez et al, 2016), yet untagged GA34 and GA69 increased the number of DNA damage-inducing R-loops in human fibroblast cell lines (Walker et al, 2017) and GA175-GFP induced phosphorylated Ataxia Telangiectasia Mutated and heterogeneous ribonucleoprotein A3 sequestration in fibroblasts and induced pluripotent stem cell-derived neurons (Nihei et al, 2020). However, it will be important to discover whether untagged polyGA can trigger this response in mammals. Despite C-terminal FLAG-tagged GA100 mimicking the aggregation profile of untagged GA100, and p62 up-regulation and DNA damage accumulation, GA100 shortened lifespan to a greater extent than GA100-FLAG. We speculate that activation of the degradation machinery, such as the proteasome or chaperones, may differentially contribute to the long-term reduction in toxicity between GA100 and GA100-FLAG.

In conclusion, we established that the adult toxicity of mCherry-tagged GR36 and PR36 was mildly increased compared with their untagged counterparts, yet tags, especially large fluorescent tags, vastly decreased polyGA toxicity and led to major structural and degradation changes of GA100. Developing new preclinical *C9orf72* mutation models where the efficacy of disease-modifying therapies can be tested would be important, and we propose the use of untagged models to facilitate the translatability of upcoming findings.

# Materials and Methods

### Husbandry of *Drosophila* stocks

*Drosophila* stocks were fed a sugar/yeast/agar (SYA) diet (Bass et al, 2007) and maintained at 65% humidity on a 12:12 h light:dark cycle. For experiments using the inducible pan-neuronal elav-GS driver, flies were maintained at 25°C, reared at controlled larval densities and allowed to mate for 48 h. Experimental flies were sorted to SYA food with 200 µM RU486 (Mifepristone) dissolved in EtOH or the same amount of EtOH-only. In experiments where construct expression was induced for 24 h, flies were sorted to EtOH-only food for 24 h, after which they were transferred to RU486 food for the indicated times. Female flies were used for all experiments and were maintained in plastic vials at a density of 12–20 flies per vial for MSD immunoassays, 15 flies per vial for phenotyping and brain-staining experiments or at 20 flies per vial for mRNA or protein isolation. Except for UAS-Proteasome subunit α5 RNAi (UAS-Pros α5 RNAi), all transgenic fly lines were backcrossed for at least six generations into the outbred WT white Dahomey strain (Grönke et al, 2010).

The elav-GS driver line was obtained as a generous gift from Dr. Hervé Tricoire (CNRS) (Osterwalder et al, 2001). The GMR-Gal4 stock was obtained from the Bloomington *Drosophila* Stock Center. The UAS-atg5 RNAi line was obtained from the Kyoto *Drosophila* Genetic Resource Center (Scott et al, 2004; Ren et al, 2009). The UAS-Pros α5 RNAi line (#16105) was obtained from the Vienna *Drosophila* Resource Center. The UAS-GA36, UAS-GA100, UAS-GR36, UAS-GR100, UAS-PR36, and UAS-PR100 (Mizielinska et al, 2014), and the UAS-GA36-mCherry, UAS-GA100-mCherry, UAS-GR36-mCherry, UAS-GR100-mCherry, UAS-PR36-mCherry, UAS-PR100-mCherry, and UAS-mCherry (Morón-Oset et al, 2019) transgenes were inserted at the attP40 docking site and used for the experiments shown in Figs 1, 2, and S1–S3. The UAS-GA100 and UAS-GA100-mCherry (Morón-Oset et al, 2019), and the UAS-GA100-GFP and UAS-GA100-FLAG (this study) transgenes were inserted at the attP2 docking site and used for experiments shown in Figs 3–5 and S4–S8.

### Generation of transgenic fly lines

To generate GFP-tagged GA100, we first PCR-amplified GFP using primers JOL124 and JOL125 and Phusion polymerase (NEB). This resulted in the addition of a NotI restriction site (RS) and the linker GGTAGTGGAAGTGGTAGT at the N-terminus of GFP, and a C-terminal KpnI RS after the stop codon. The linker encoded 3xglycine–serine

**Table 1. List of primers.**

| Method | Primer name | Primer sequence | Purpose |
|---|---|---|---|
| Cloning | JOL26 | ATATGAATTCGGATCCCACCATG | Generation of the pUAST GA100FLAG plasmid |
| | JOL117 | AAAAGCGGCCGCTTACTTATCGTCGTCGTCCTTGTAATCTGCTCCTGCT | Generation of the pUAST GA100FLAG plasmid |
| | JOL124 | ATATGCGGCCGCCGGTAGTGGAAGTGGTAGTATGGTGAGCAAGGGCGAGGAGCTGTTCAC | Generation of the pUAS T-GFP-C plasmid |
| | JOL125 | AAAAGGTACCTCACTTGTACAGCTCGTCCATGCGGAGAGTGAT | Generation of pUAS T-GFP-C and GFP-only pUAST plasmids |
| | JOL126 | ATATGAATTCCAACATGGTGAGCAAGGGCGAGGAG | Generation of the GFP-only pUAST plasmids |
| qRT–PCR | JOL267 | GTACGACAGAGGCGTGAACA | qRT–PCR for Pros α5 |
| | JOL268 | CCACCTCCACAATCTTCTCC | |

(Gly-Ser) (Morón-Oset et al, 2019). After digestion of the amplicon, it was ligated into the pUAST attB vector to form the pUAST-GFP-C plasmid. Then, the pBlueScript SK(+)-EcoRI-ATG-GA100-NotI plasmid (Morón-Oset et al, 2019) was digested and subsequently ligated into the pUAST-GFP-C plasmid. As a control, we also PCR-amplified GFP using primers JOL125 and JOL126, which allowed for the addition of an N-terminal EcoRI RS and an ATG initiation site, and a C-terminal NotI RS after the stop codon. To create GA100-FLAG, we PCR-amplified the sequence for GA100 using JOL26 and JOL117, the latter containing the FLAG-coding sequence followed by a stop codon and a NotI RS. No linker was included between GA100 and FLAG for this plasmid. This amplicon was then directly ligated into the pUAST attB plasmid.

The sequences of all primers are included in Table 1. To achieve high expression levels, all constructs contained a CACC Kozak sequence before the ATG start codon. The sequence of all plasmids was verified by Sanger sequencing (Eurofins Genomics). Constructs were inserted into the fly genome using phiC31-mediated attP/attB site-directed integration (Bischof et al, 2007). Plasmids were injected by the BestGene *Drosophila* Embryo Injection Service.

### Egg-to-adult viability assay and eye phenotypes

Five virgin GMR-Gal4 females were mated with five UAS or WT males for 2 d, then transferred to experimental vials and allowed to lay eggs for 5 h at 25°C on SYA food. Eggs were counted and vials were incubated at 25°C or 29°C to achieve mid or high-transgene expression levels, respectively, as the Gal4-UAS system is temperature-sensitive (Duffy, 2002). The number of eclosed adult flies was counted. Egg-to-adult viability was calculated by dividing the number of adult flies by the number of eggs. 10 replicates per genotype and temperature were used. Images of fly eyes were taken on the day of emergence using a Leica M165 FC stereomicroscope equipped with a motorized stage and a multifocus tool (Leica application suite software). Eye area was calculated using ImageJ from 12 fly eyes. Fly eyes were scored in a blinded manner.

### Lifespan assay

Flies were sorted into experimental vials at a density of 15 flies per vial containing SYA medium with EtOH-only or with 200 μM RU486 to induce transgene expression. 10 independent biological replicates per condition were tested (i.e., n = 150 female flies per genotype and treatment). Flies were tipped to fresh vials 2–3 times per week and,

at the same time, deaths were scored. Data are shown as cumulative survival curves.

### RNA extraction, cDNA synthesis, and quantitative real-time PCR (qRT–PCR)

Total RNA was extracted using Trizol (Invitrogen) according to the manufacturer's instructions. RNA was treated with DNase I (Thermo Fischer Scientific) and RNA concentration was measured using the Qubit BR RNA assay (Thermo Fisher Scientific). cDNA was generated from 600 ng total RNA using the SuperScript III first-strand synthesis kit (Invitrogen) and oligodT primers, according to the manufacturer's instructions. qRT–PCR was conducted on a QuantStudio7 (Thermo Fisher Scientific) using the PowerUp SYBR Green Master Mix (Thermo Fisher Scientific). Relative gene expression (fold induction) was calculated using the ΔΔCT method and Rpl32 as a normalization control.

### Western blotting

20 adult fly heads were homogenized in 100 μl of ice-cold RIPA supplemented with cOmplete MINI without EDTA protease inhibitor (Roche) and PhosStop phosphatase inhibitors (Roche), and incubated on ice for 30 min with occasional vortexing. Samples were then centrifuged at 13,000$g$ for 15 min at 4°C, after which the supernatant was retrieved. Protein samples were pipetted up and down several times before loading the RIPA protein samples to prevent or minimize the loss of aggregated proteins. 15 μl per sample were separated on any-kD stain-free Criterion gels (Biored) and subsequently transferred to 0.45 μm nitrocellulose membranes (GE Healthcare). Protein loading was imaged by exposing membranes to UV light. Membranes were subsequently blocked with 5% non-fat dry milk for 1 h at RT and incubated overnight at 4°C with a mouse anti-GA (clone 5E9) (1:1,000; AB_2728663; Merck Millipore), rabbit anti-mCherry (1:1,000, AB_2571870; Abcam), HRP-conjugated anti-GFP (1:1,000, AB_247003; Miltenyi Biotec), and rabbit anti-Ref(2)P/p62 (1:1,000; catalog #178440; Abcam). After three washes in TBST, membranes were probed with HRP-conjugated anti-mouse (1:10,000, AB_2536527; Thermo Fisher Scientific) or anti-rabbit (1:10,000, AB_2536530; Thermo Fisher Scientific) secondary antibodies for 1 h at RT and detection was performed using an ECL chemiluminescence kit (GE Healthcare). ImageJ was used for band intensity quantifications.

### DPR MSD immunoassays

Adult flies pan-neuronally expressing arginine-rich DPRs or GA100 were fed food containing RU486 for 3 or 5 d, respectively. 12–20 female fly heads per replicate and genotype were used. Flies were frozen in liquid nitrogen and heads removed. GR and PR levels were measured according to Moens et al (2018). Therefore, fly heads were homogenised in 100 $\mu$l of RIPA buffer with 2% SDS buffer, sonicated in a water bath at 60% Amp for 45 s (1 s on – 1 s off cycles) and centrifuged at 13,000$g$ for 20 min at RT. The supernatant was collected in fresh tubes and protein concentration was determined by DC protein assay (Bio-Rad). Soluble and insoluble GA levels were measured according to Quaegebeur et al (2020). Therefore, fly heads were homogenized in 100 $\mu$l of RIPA with 2% SDS and centrifuged at 2,000$g$ to clear debris. Clean supernatant was transferred into ultracentrifuge tubes and used for BCA. Lysates were centrifuged at 100,000$g$ for 30 min at 16°C to obtain the soluble fraction. Pellets were resuspended in 30 $\mu$l 7M urea, which were sonicated in a water bath until dissolved. The samples were then ultracentrifuged at 100,000$g$ for 30 min at 16°C and the supernatant was collected to obtain the insoluble fraction. The samples were diluted to 1M urea with TBS with PI tablets.

MSD immunoassay was performed in single-plex using 96-well SECTOR plates (MSD) to quantify polyGA, polyGR, and polyPR expression levels. Assays were performed as previously described (Simone et al, 2018). Plates were coated with unlabelled anti-polyGA, anti-polyGR and anti-polyPR antibodies. After blocking with 3% milk, samples were loaded and incubated overnight at 4°C. The volume loaded per well for the GA assay was adjusted to 25 $\mu$g of soluble fractions and 4.5 times that amount for insoluble fractions, whereas 77 $\mu$g of total protein were loaded per well for the GR and PR assays. Freshly extracted samples were run in duplicate, avoiding freeze–thaw. A four-parameter logistic regression curve was fit to the values obtained from a standard curve of peptide calibrators using GraphPad Prism, and concentrations were interpolated. The following antibodies were used: anti-GA (clone 5E9, AB_2728663; Merck Millipore) as capture antibody and biotinylated, streptavidin-conjugated, sulfo-tagged anti-GA (GA5F2, provided by Prof. Dr. Edbauer, Ludwig-Maximilians-Universität, München) as detector antibodies; anti-GR (GR661, custom-made from Eurogentec) as captured and biotinylated, streptavidin-conjugated, sulfo-tagged anti-GR (AB_2728664, in-house) as detector antibodies; and anti-PR (non-biotinylated, and biotinylated, streptavidin-conjugated, and sulfo-tagged PR32B3, provided by Prof. Dr. Edbauer, Ludwig-Maximilians-Universität, München) for both capture and detector antibodies. Plates were read with the MSD reading buffer (R92TC; MSD) using the MSD Sector Imager 2400. Signals correspond to the intensity of emitted light upon electrochemical stimulation of the assay plate. Before analysis, the average reading from a calibrator containing no peptide was subtracted from each reading.

### Pulse-and-chase assay

For Western blotting analysis, flies were fed on RU486 food for 25 d, then frozen and used as the pulse. Alternatively, 25-d-old flies fed on RU486 were tipped onto EtOH food for 4 d, then frozen and used

as the chase. Three biological replicates of female flies containing 20 heads were used per genotype and treatment. For brain immunostainings, two batches of flies were used. Flies used for the chase were sorted first, maintained on RU486 food for 25 d and then transferred to EtOH food for 4 d. Flies used for the pulse were sorted 4 d later, and maintained on RU486 food for 25 d. Fly brains of both batches were dissected, stained, and imaged on the same day to reduce batch effects.

### Histology of adult fly brains and brain immunostainings

Brains of adult female flies were dissected in PBS and immediately fixed in 4% paraformaldehyde at 4°C for 2 h as previously described (Morón-Oset et al, 2019). Brains were washed in PBS with 0.5% Triton X-100 (PBT) at RT and blocked in PBT with 5% fetal bovine serum and 0.01% sodium azide for 1 h at RT, then incubated with mouse monoclonal anti-GA (1:3,000; AB_2728663; Merck Millipore), 5H9 rat anti-polyGR (1:50 [Mori et al, 2013]) or rabbit polyclonal anti-PR (1:1,000; catalog #23979-1-AP; Proteintech) antibodies overnight at 4°C. After washes in PBT at RT, brains were incubated overnight at 4°C at 1:1,000 dilution with Alexa488 goat anti mouse (catalog #A11001; Thermo Fisher Scientific), Alexa633 goat anti-mouse (AB_2535718; ; Thermo Fisher Scientific), Alexa488 goat anti-rabbit (catalog #A11008; Thermo Fisher Scientific) or Alexa488 goat anti rat (catalog #A11006; Thermo Fisher Scientific). Finally, brains were washed in PBT, incubated in glycerol-PBS, and mounted in VectaShield antifade mounting medium with DAPI (catalog #H-1200; Vectorlabs).

### TUNEL assay

Dissected fly brains of adult female flies were fixed in 4% paraformaldehyde at 4°C for 2 h, washed in PBS with 0.5% Triton X-100 at RT and incubated in a solution containing the TUNEL enzyme and the TUNEL label for 1 h at 37°C under agitation. The brains were then washed in PBT 2 × 30 min, incubated in glycerol-PBS and mounted in VectaShield antifade-mounting medium with DAPI (catalog #H-1200; Vectorlabs).

### Imaging of adult *Drosophila* brains and quantification

Using a Leica SP8-DLS confocal microscope, series of 2-$\mu$m z-stacks across the whole fly brain were taken, with the same settings used across genotypes. Brains were imaged using a 20X objective. To maximize the detectability of specific signal, HyD detectors, gating and the excitation wavelength that maximized the fluorescence emission of all fluorophores were used during imaging. GR and PR levels were quantified in the pars intercerebrallis region using maximum intensity projections from z-stacks in ImageJ. The same settings were used for all conditions of a given experiment. To assess changes in TUNEL-positive cells, whole stacks were taken and quantified in 3D using the image analysis software Imaris 9.2.0 (Oxford Instruments). After background correction, the built-in spot detection algorithm was used to identify spots with a minimum size of 1,000 nm. Detection settings were adjusted based on the maximum intensity of the spots, which proved the most accurate filter to distinguish between strongly labelled spots (considered as real TUNEL puncta) and weak/low quality spots from trachea or

background. To assess changes in GA- and mCherry-positive aggregates, we used Imaris 9.2.0 (Oxford Instruments) and the built-in surface detection algorithm to identify aggregates with a grain size above 5 nm. Detection settings were adjusted based on the maximum intensity of the surfaces. The same parameters were used for all of the conditions compared in the same experiment.

## Statistics

Statistical analysis was performed using GraphPad Prism or RStudio version 4.0.4. Individual statistical tests are indicated in the figure legends. For multiple comparison testing of parameters other than lifespan, one-way and two-way ANOVA were used. Bonferroni posthoc test was used when two groups were compared. Tukey–Kramer test was used when more than two groups were compared. Familywise error rate was controlled using the Šidák's multiple corrections test. Lifespan comparisons across genotypes of the same treatment were performed using RStudio version 4.0.4. Pairwise comparisons were assessed using log-rank with Bonferroni's multiple-testing correction. $P$-values < 0.05 were considered significant: $*P < 0.05$, $**P < 0.01$, $***P < 0.001$, and $****P < 0.0001$.

## Supplementary Information

## Acknowledgements

Imaging analyses were performed in the FACS & Imaging Core Facility at the Max Planck Institute for Biology of Ageing. Stocks obtained from the Bloomington *Drosophila* Stock Center (NIH P40OD018537), the Kyoto *Drosophila* Genetic Resource Center, and the Vienna *Drosophila* Resource Center were used in this study. Javier Morón-Oset received support by the Cologne Graduate School of Ageing Research, which is funded by the Max Planck Society and the Deutsche Forschungsgemeinschaft (DFG). We thank Prof. Dr. Edbauer (Ludwig-Maximilians-Universität, München) for kindly sharing his GA5F2 and PR32B3 antibodies. This work was funded by the Max Planck Society, the Wellcome Trust (WT098565/Z/12/Z), and Alzheimer's Research UK and the UK Dementia Research Institute, which receives its funding from UK DRI, funded by the UK Medical Research Council, Alzheimer's Society, and Alzheimer's Research UK.

### Author Contributions

J Morón-Oset: conceptualization, data curation, investigation, and writing—original draft.
LKS Fischer: data curation and formal analysis.
M Carcolé: data curation and methodology.
A Giblin: data curation and methodology.
P Zhang: data curation.
AM Isaacs: conceptualization and writing—original draft.
S Grönke: conceptualization, investigation, and writing—original draft.
L Partridge: conceptualization, supervision, funding acquisition, investigation, and writing—original draft, review, and editing.

## Conflict of Interest Statement

The authors declare that they have no conflict of interest.

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
