## [Reviewer comments · Life Science Alliance]

Life Science Alliance

Toxicity of C9orf72-associated dipeptide repeat peptides is modified by commonly used protein tags

Javier Morón-Oset, Lilly Fischer, Mireia Carcolé, Ashling Giblin, Pingze Zhang, Adrian Isaacs, Sebastian Grönke, and Dame Partridge

DOI: <https://doi.org/10.26508/lsa.202201739>

Corresponding author(s): Dame Partridge, Max Planck Institute for Biology of Ageing

Review Timeline:

Submission Date:	2022-09-27
Editorial Decision:	2022-11-18
Revision Received:	2023-04-30
Editorial Decision:	2023-05-26
Revision Received:	2023-06-01
Accepted:	2023-06-03

Scientific Editor: Novella Guidi

Transaction Report:

November 18, 2022

Re: Life Science Alliance manuscript #LSA-2022-01739

Prof. Linda Partridge
Max Planck Institute for Biology of Ageing
Biological Mechanisms of Ageing
Joseph-Stelzmann Str, 9b
Cologne 50931
Germany

Dear Dr. Partridge,

Thank you for submitting your manuscript entitled "Toxicity of C9orf72-associated dipeptide repeat peptides is modified by commonly used protein tags" to Life Science Alliance. The manuscript was assessed by expert reviewers, whose comments are appended to this letter. We invite you to submit a revised manuscript addressing the Reviewer comments.

Thank you for this interesting contribution to Life Science Alliance. We are looking forward to receiving your revised manuscript.

Sincerely,

B. MANUSCRIPT ORGANIZATION AND FORMATTING:

Reviewer #1 (Comments to the Authors (Required)):

In this manuscript, Morón-Oset et al. performed in vivo study of poly-GA, poly-GR and poly-PR, the toxic DPRs that are RAN-translated from C9orf72 GGGGCC.GGCCCC RNA repeats. The authors comprehensively investigated the effect of different tags (fluorescent and epitope tags that are commonly used in biological research) on DPR toxicity using Fly model, showing that mCherry tag increased the toxicity of arginine-enriched DPRs, but reduced toxicity of poly-GA100. Interestingly, they show that other tags such as GFP and FLAG also reduced GA100 toxicity. Relevant to this, the authors found that fluorescent tags could affect GA100 aggregation and degradation, as well as GA-mediated DNA damage and p62 regulation.

Overall, the authors have conducted a well-designed and suggestive study that is of sufficient interest to the to the C9orf72-ALS/FTD field, especially the researchers who are interested in the study of gain of function hypothesis in association with C9 RAN translation. Although there are some issues, this manuscript is overall promising and fully studied the unexpected but important effects of the tags on C9orf72 DPR toxicity.

Before publication, the following points need to be addressed by the authors:

1. It is very convincing to see that mCherry tag can enhance the toxicity of poly-GR and poly-PR, two most toxic C9 DPRs, but one may wonder whether it is directly caused by the tag itself, or indirectly caused by the altered cellular localization - nucleolar dysfunction has been linked to poly-GR and poly-PR toxicity - is that possible mCherry tag affects the nucleolar localization of poly-GR and poly-PR? This part is crucial since it is associated with the mechanism underlying arginine-containing DPR toxicity, which is lacking in the present study and need to be included after revision, at least in the Discussion section.
2. How to maintain the similar protein levels of mCherry-tagged or untagged poly-GR/poly-PR, which is helpful for the comparison of toxicity?
3. It is good to see mCherry and GFP tags affect the turnover and aggregation of poly-GA100. Since it is possible that longer GA repeats can be degraded by both autophagy (cytosolic) and the proteasomes (cytosolic and nuclear), it would be nice to perform biochemical assays to see whether the cytosolic and nuclear fractions of poly-GA100 is affected by the tags.

Reviewer #2 (Comments to the Authors (Required)):

In this manuscript Morón-Oset et al., use Drosophila models expressing c9orf72 related dipeptide repeat proteins (DPRs) to demonstrate that addition of commonly used tags (GFP, mCherry and FLAG) may alter some commonly observed phenotypes. The study is limited to the use of the PolyGA, PolyPR and PolyGR DPRs at 36 and 100 repeats in length. While the effect of tags upon DPRs is an important question there are several major issues with the manuscript that need to be addressed.

Major concerns

1. A major concern with the study is the lack of evidence demonstrating that the differences in toxicity seen between tagged and untagged lines is not due to different expression levels between the lines used.
 - In the methods the authors state that most of the lines they are using are landed in the attP40 site whilst UAS-GA100 and UAS-mCherry lines are at the attP2 site and there is no data given for where the new GFP and FLAG lines are landed. It is critical that the authors are explicitly clear which lines are landed in which site for each experiment as lines in different sites cannot be reliably compared. This is especially important given evidence that insertion at Attp40 site can have detrimental effects upon viability (<https://doi.org/10.1101/2022.05.14.491875>).
 - In Figure 3A the authors show that the amount of GA translated from each line is highly variable with much greater expression levels from the tagged lines (particularly GFP and mCherry). This raises significant questions about the expression levels in the

GR and PR lines. If these lines were made the same way and tagged GR and PR lines also show much higher expression levels this may be the cause of the greater toxicity in the GR and PR experiments. It is critical that the authors provide quantified blots for every line used in these experiments so we can be sure the effects are not due to variable expression levels.

2. In the introduction the authors state that previous studies show that GFP tagging is not detrimental to long 400 (murine) and 1000 (fly) repeat models but they are to shorter 100 and 175 repeat models. This is a critical point suggesting that toxicity may be directly related to the relative length of the DPR and the tag. In shorter repeat models the tag is a greater proportion of the whole protein, so may have a more significant impact. This is supported by the authors own data showing that the shorter FLAG tags have less impact on the DPR phenotypes. The authors should examine the effect upon more disease relevant repeat lengths (>400 repeats) to establish whether only short repeat models are affected by the tags.

3. The data presented in figure 3 are not entirely convincing due to the reliance on the observation of high-molecular weight bands as a readout of aggregation of insoluble DPRs. This data should be validated by performing fractionation of the soluble and insoluble fractions and quantification using meso scale discovery ELISA assays as described in the co-authors previous work (<https://doi.org/10.1186/s40478-020-01036-y>). Immunohistochemical analysis of fly brains should also be used to examine the number of DPR aggregates.

4. Following on from the previous point, the authors should examine the effect of the tags upon localisation and morphology of the DPR "aggregates".

5. None of the eye data (Fig1A-B, sup fig 1A-B) is quantified. One could argue that in both figures the GR100-mcherry eye looks less severely impacted than the untagged GR100, suggesting a very mild rescue. All eye data needs to be quantified using an appropriate scoring system. Given the multifaceted nature of these phenotypes a multi-point scoring system such as described by Pandey et al (2007) (<https://doi.org/10.1038/nature05853>) should be used.

Minor concerns

1. Control data in egg-to-adult viability assays is repeated between graphs. E.g. the data shown for WT and mCherry controls in figure 1C is identical to figure 1D. Data should not be re-used for independent experiments. If this is one experiment it should be graphed together. This is the same for the supplementary data.

2. Western blots should be repeated with anti-GFP/anti-mcherry/anti-FLAG to determine whether the tags remain attached or have been cleaved from the DPR. Cleavage of tags, especially GFP and mCherry has been commonly observed.

3. The landing site for the new lines generated in this study needs to be included in the methods.

Reviewer #3 (Comments to the Authors (Required)):

The toxicity of RAN translation products of the C9ORF72 expanded repeat is typically demonstrated using exogenously expressed proteins of a selected reading frame and with addition of a fluorescent protein or small epitope tag. Morón-Oset et al. use *Drosophila* to show that addition of either a fluorescent protein or an epitope tag can impact the toxicity of the RAN protein product. The authors first used expression of GA, GR or PR with 36 or 100 repeats and with or without a C-terminal mCherry tag in fly eye. They also used higher and lower expression levels. GA showed no effects between tagged and untagged but GR and PR dipeptide repeats (DPRs) disrupted eye morphology and reduced egg to adult survival and, importantly, the effects were stronger with a C-terminal mCherry tag. The results show that toxicity with regard to survival can be maxed out with 100 repeats and the impact of mCherry is not observed but mCherry particularly reduces survival upon expression of PR DPRs. A test of expressed DPR in adult neurons demonstrated that a C-terminal mCherry rescued the decreased life span observed with 36 or 100 GA DPRs. Overall the mCherry tag increased toxicity of arginine DPRs and reduced toxicity of GA DPRs in vivo. Their results also show that different tags have different effects on in vivo toxicity. GFP and mCherry completely blocked lifespan shortening of GA DPRs and the short Flag tag reduced but did not eliminate lifespan shortening. Overall there was a clear modifier effect of a fluorescent or small epitope tag. Importantly the authors show that differences on in vivo toxicity does not correlate with levels of protein expression but do correlate with propensity to aggregate into high molecular weight species on a denaturing gel. Using p62 upregulation as a measure of cell stress following induced expression of different DPR forms in neurons, the authors show that large fluorescent protein tags but not a FLAG tag, prevented p62 upregulation by GA DPRs. Similarly a TUNEL assay showed that GA DPRs with and without Flag increased DNA fragmentation but an mCherry tag prevented this effect.

The main finding that tags on DPRs impacts toxicity is an important point and the authors present evidence that aggregation, p62 upregulation for GA DPRs and toxicity due to DNA damage is involved. The role of aggregation is strongly suggestive but not definitive since it is based on a signal on western blots what appears to be just below the bottom of the well. One concern is that the protein samples were centrifuged prior to loading the supernatant so that aggregated proteins could be lost prior to the western and comparisons only on the westerns might not reflect differences in aggregation. Another somewhat minor concern is the transition through the manuscript of testing multiple DPRs to only GA DPRs. The results address the potential basis for

discrepancies regarding GA DPR toxicity, which is important. There are different effects of the same tags on different DPRs that are not addressed mechanistically. The mechanisms of these effects are of interest but not critical.

Thank you for the reviewers' comments on our submission to Life Science Alliance. We were delighted to see that all three reviewers found our study interesting and acknowledged the importance of this study in understanding the impact of protein tags on the toxicity of C9orf72 dipeptide repeat proteins (DPRs) for developing more accurate animal models for ALS and FTD.

We are grateful to the reviewers for their constructive criticism and based on their comments we have conducted several new experiments that strengthen our findings. This includes, the quantification of the protein levels of tagged and untagged DPRs by immunohistochemistry and meso scale discovery immunoassays as requested by reviewers 1 and 2. We also used immunohistochemistry on fly brains to show that the protein tags do not interfere with the subcellular localisation or aggregate morphology of the DPRs. Furthermore, we performed new experiments that suggest that tagged GA protein is preferentially degraded by the proteasome and performed additional western blots with tag specific antibodies as requested by reviewer 2. We have also updated the material and method section to include more precise information about the genomic landing sites to generate the transgenic fly lines and quantified the eye phenotypes. In summary, we have addressed all reviewer comments and we are confident that the revised manuscript is now suitable for publication in Life Science Alliance. Please find below a point-by-point response to the reviewers' comments.

Reviewer 1 (Comments to the Authors (Required)):

In this manuscript, Morón-Oset et al. performed in vivo study of poly-GA, poly-GR and poly-PR, the toxic DPRs that are RAN-translated form C9orf72 GGGGCC.GGCCCC RNA repeats. The authors comprehensively investigated the effect of different tags (fluorescent and epitope tags that are commonly used in biological research) on DPR toxicity using Fly model, showing that mCherry tag increased the toxicity of arginine-enriched DPRs, but reduced toxicity of poly-GA100. Interestingly, they show that other tags such as GFP and FLAG also reduced GA100 toxicity. Relevant to this, the authors found that fluorescent tags could affect GA100 aggregation and degradation, as well as GA-mediated DNA damage and p62 regulation.

Overall, the authors have conducted a well-designed and suggestive study that is of sufficient interest to the to the C9orf72-ALS/FTD field, especially the researchers who are interested in the study of gain of function hypothesis in association with C9 RAN translation. Although there are some issues, this manuscript is overall promising and fully studied the unexpected but important effects of the tags on C9orf72 DPR toxicity.

Before publication, the following points need to be addressed by the authors:

1. It is very convincing to see that mCherry tag can enhance the toxicity of poly-GR and poly-PR, two most toxic C9 DPRs, but one may wonder whether it is directly caused by the tag itself, or indirectly caused by the altered cellular localization - nucleolar dysfunction has been linked to poly-GR and poly-PR toxicity - is that possible mCherry tag affects the nucleolar localization of poly-GR and poly-PR? This part is crucial since it is associated with the mechanism underlying arginine-containing DPR toxicity, which is lacking in the present study and need to be included after revision, at least in the Discussion section.

We thank the reviewer for this important suggestion. In order to address whether the addition of the tag alters the subcellular localization of poly-GR and poly-PR, we performed immunostainings on fly brains using GR- and PR-specific antibodies after 3 days of pan-neuronal transgene induction (**Fig 2A-B & Supplementary Fig S3A-F**). The following transgenes were included in the analysis: GR36, GR100, PR36 and PR100 with and without an mCherry-tag. For poly-GR, the GR signal was almost exclusively detected in the insulinergic Median Neurosecretory Cell (MNC) region (**Supplementary Fig S3A, C**) (de Velasco et al. 2007), as previously reported (Morón-Oset et al. 2019). The pattern of the GR signal in all genotypes was exclusively nuclear and diffuse, and no obvious morphological differences were observed between untagged and mCherry-tagged polyGR proteins of different repeat lengths (**Fig 2A & Supplementary Fig S3E**). The polyPR signal was more widely detected than polyGR and, similar to polyGR, it also accumulated in MNCs, on which we focused for further morphological assessment (**Fig 2B & Supplementary Fig S3F**). PR36 showed an exclusively nuclear and diffuse localization, whereas the PR100 signal was rather punctated and accumulated both inside and outside of the nucleus (**Fig 2B & Supplementary Fig S3F**). While protein length affected the subcellular localization of PR, mCherry-tagging did not overtly influence the morphology of PR36 or PR100 (**Fig 2B**). Thus, in summary, our results suggest that the addition of the mCherry-tag does not affect the subcellular localization of poly-GR and poly-PR. These data have now been included in the manuscript as **Fig 2** and **Supplementary Fig S3**.

2. How to maintain the similar protein levels of mCherry-tagged or untagged poly-GR/poly-PR, which is helpful for the comparison of toxicity?

In order to address whether differences in protein levels of mCherry-tagged and untagged poly-GR/poly-PR might contribute to the observed difference in toxicity, we quantified GR and PR protein expression by immunostaining in the pars intercerebralis region of the fly brains, where these proteins can easily be detected (this was done on the same fly brains used for the subcellular localization analysis in point 1, **Fig 2** and **Supplementary Fig S3**). To complement these measurements, we used Meso Scale Discovery (MSDs) immunoassays to address GR and PR levels in whole fly heads. We did not detect any significant differences in GR expression levels between mCherry-tagged and untagged GR by immunostaining (**Fig 2C**) or by MSD (**Fig 2D**), suggesting that the differences in toxicity between the tagged and untagged protein cannot simply be explained by a change in protein levels. In contrast, the brains of PR36-mCherry-expressing flies showed a stronger PR signal than those of untagged PR36 flies (**Fig 2E**). In agreement, we also found a non-significant trend towards increased PR signal in PR100-mCherry flies compared to PR100 flies (**Fig 2E**). Unfortunately, we were unable to confirm this finding by MSD, as the detection did not work reliably for PR (**Supplementary Fig S3G**). In summary, while GR levels were not changed upon addition of the mCherry tag, PR levels were increased, which might contribute to the higher toxicity of this DPR. The new data have been added to the manuscript (**Fig 2** and **Supplementary Fig S3**) and the findings are also included in the discussion.

3. It is good to see mCherry and GFP tags affect the turnover and aggregation of poly-GA100. Since it is possible that longer GA repeats can be degraded by both autophagy (cytosolic) and the proteasomes (cytosolic and nuclear), it would be nice to perform biochemical assays to see whether the cytosolic and nuclear fractions of poly-GA100 is affected by the tags.

While we agree with the reviewer that it is an interesting question whether tagged poly-GA proteins are degraded by autophagy or the proteasome. In our experience, subcellular fractionation of aggregation-prone proteins from fly heads is not trivial. Thus, we chose to address the question with two different approaches: (I) Analysis of the subcellular localization of tagged GA100 by immunohistochemistry (**Supplementary Fig S5**) and (II) measurement of GA levels in flies co-expression tagged GA100 with either RNAi against ATG5 (to downregulate autophagy) or Pros $\alpha 5$ RNAi (to downregulate proteasome activity) (**Supplementary Fig S8**).

Immunostainings of fly brains expressing untagged GA100 and GA100-FLAG showed that GA aggregates were located exclusively in the cytosol, indicated by the lack of an overlap with the nuclear DAPI counterstain (**Supplementary Fig S5B**). The brains of GA100-GFP and GA100-mCherry flies showed GFP- and mCherry-positive aggregates (**Supplementary Fig S5A**), but these only showed a diffuse cytosolic GA-signal that did not co-localize with the GFP or mCherry fluorescent signals, respectively (**Supplementary Fig S5A, B**). The diffuse GA signals may indicate degradation products of GA, in agreement with our western blot results (**Supplementary Fig S6A, H**). The high density of GFP- and mCherry-positive inclusions in the brains of GA100-GFP and GA100-mCherry flies may explain the limited penetration of our anti-GA antibody (**Supplementary Fig S5A, B**), which was also previously observed in cells using a different anti-GA antibody (Liu et al. 2022). Importantly, no nuclear GA or fluorescent tag-specific signals were detected in the brains of GA100-GFP or GA100-mCherry flies (**Supplementary Fig S5B**),

indicating that the subcellular localization of these constructs is exclusively cytoplasmic in fly brains regardless of the presence or absence of protein tags, consistent with previous findings (Mizielinska et al. 2017; Morón-Oset et al. 2019).

To further investigate whether the proteasome or autophagy play a role in the increased degradation propensity of GA100 containing fluorescent tags, we focused on GA100-mCherry and generated flies that pan-neuronally co-expressed GA100-mCherry in combination with either RNAi against the autophagy-regulator atg5 (atg5 RNAi), which was previously shown to block autophagic flux (Bjedov et al. 2020; Lu et al. 2021; Juricic et al. 2022), or RNAi against the proteasome subunit $\alpha 5$ (Pros $\alpha 5$ RNAi). Pros $\alpha 5$ is a proteasome subunit of the 20S catalytic core required for the hydrolytic activity of the 26S proteasome machinery, which is key for tissue integrity in *Drosophila* (Velentzas et al. 2013). Pros $\alpha 5$ RNAi for 5 days reduced Pros $\alpha 5$ transcript levels (**Supplementary Fig S7A**) and strongly increased the protein levels of p62 (**Supplementary Fig S7B, C**), a common response to proteasome inhibition to enable the autophagic clearance of aggregated proteins (Nezis et al. 2008; Velentzas et al. 2013; Péter et al. 2013), consistent with downregulation of proteasome function by the Pros $\alpha 5$ RNAi.

Blocking autophagy or the proteasome might lead to an accumulation of the tagged protein, as it cannot be efficiently degraded anymore. Therefore, we next quantified GA100-mCherry levels via immunohistochemistry on fly brains using an anti-GA antibody (**Supplementary Fig S8**). Interestingly, compared to flies expressing GA100-mCherry-alone, GA levels were increased upon co-expression with Pros $\alpha 5$ RNAi (**Supplementary Fig S8A, B**), and this effect was also significant when the mCherry signal was used for quantification (**Supplementary Fig S8A, C**). There was also a very mild trend for increased GA levels upon co-expression with atg5 RNAi. However, this did not reach significance, neither upon quantification of the GA or mCherry signals (**Supplementary Fig S8A-C**). Overall, our data suggest that mainly the proteasome contributes to the degradation of GA100-mCherry, while autophagy only seems to play a minor role. Consistently, GFP-tagged polyGA has been shown to bind to the proteasome (Guo et al. 2018; Khosravi et al. 2020). These new data have been added to the manuscript in **Supplementary Fig S5, S7, S8** and are also included in the discussion.

Reviewer 2 (Comments to the Authors (Required)):

In this manuscript Morón-Oset et al., use *Drosophila* models expressing c9orf72 related dipeptide repeat proteins (DPRs) to demonstrate that addition of commonly used tags (GFP, mCherry and FLAG) may alter some commonly observed phenotypes. The study is limited to the use of the PolyGA, PolyPR and PolyGR DPRs at 36 and 100 repeats in length. While the effect of tags upon DPRs is an important question there are several major issues with the manuscript that need to be addressed.

Major concerns

1. A major concern with the study is the lack of evidence demonstrating that the differences in toxicity seen between tagged and untagged lines is not due to different expression levels between the lines used.

• In the methods the authors state that most of the lines they are using are landed in the attP40 site whilst UAS-GA100 and UAS-mCherry lines are at the attP2 site and there is no data given for where the new GFP and FLAG lines are landed. It is critical that the authors are explicitly clear which lines are landed in which site for each experiment as lines in different sites cannot be reliably compared. This is especially important given evidence that insertion at Attp40 site can have detrimental effects upon viability (<https://doi.org/10.1101/2022.05.14.491875>).

As suggested, we have now included precise information about the landing sites used for the different experiments. Similar to (Mizielinska et al. 2014), polyGR and polyPR transgenes were inserted into the attP40 landing site on chromosome II across all experiments. In experiments where polyGA was compared to polyGR and polyPR, the polyGA transgenes were inserted into the same landing site (i.e., attP40). In experiments where the effect of different tags on GA100 was assessed, all constructs were inserted into the attP2 landing site on chromosome III, similar to (Morón-Oset et al. 2019). This has now been clarified both in the methods and in the results section.

• In Figure 3A the authors show that the amount of GA translated from each line is highly variable with much greater expression levels from the tagged lines (particularly GFP and mCherry). This raises significant questions about the expression levels in the GR and PR lines. If these lines were made the same way and tagged GR and PR lines also show much higher expression levels this may be the cause of the greater toxicity in the GR and PR experiments. It is critical that the authors provide quantified blots for every line used in these experiments so we can be sure the effects are not due to variable expression levels.

We agree with the reviewer that this is an important control. As GR and PR protein cannot be detected by western blot, we quantified the levels of GR and PR using immunohistochemistry and via Meso Scale Discovery (MSDs) immunoassays (for more details see also the reply to point 2 of reviewer 1). In summary, we observed that protein levels of GR were not changed when tagged with mCherry, indicating that the increase in toxicity cannot simply be explained by higher levels of toxic GR (**Fig 2D, E**). In contrast, we observed an increase in PR levels upon tagging with

mCherry, which may contribute to the higher toxicity of this DPR (**Supplementary Fig 3G**). The new data have been added to the manuscript (**Fig 2** and **Supplementary Fig S3**) and the findings are also included in the discussion.

2. In the introduction the authors state that previous studies show that GFP tagging is not detrimental to long 400 (murine) and 1000 (fly) repeat models but they are to shorter 100 and 175 repeat models. This is a critical point suggesting that toxicity may be directly related to the relative length of the DPR and the tag. In shorter repeat models the tag is a greater proportion of the whole protein, so may have a more significant impact. This is supported by the authors own data showing that the shorter FLAG tags have less impact on the DPR phenotypes. The authors should examine the effect upon more disease relevant repeat lengths (>400 repeats) to establish whether only short repeat models are affected by the tags.

We agree with the reviewer that it would be interesting to address how tags affect the toxicity of very long DPRs (>400 repeats). However, the generation of constructs for very long DPRs is not trivial, due to the high repetition in sequence, and it was not feasible to generate these constructs in time for the revision. As we currently have no fly lines available to directly compare the effect of tagged vs untagged DPRs for construct with more than 400 repeats, we could not directly address this point. Noteworthy, however, the repeat length used in this study falls well into the length of *C9orf72* hexanucleotide repeat expansions that can cause ALS/FTD in human patients (Dols-Icardo et al. 2014; Van Mossevelde et al. 2017), suggesting that the results presented here cover disease relevant repeat lengths. Furthermore, the exact repeat length of DPRs in patients with very long hexanucleotide repeat expansions is currently unknown, and there is no clear correlation between repeat length and disease onset or progression (van Blitterswijk et al. 2013; Gijssels et al. 2016; Fournier et al. 2019). Thus, whether only long-repeat DPRs are relevant for ALS/FTD is not clear and the point raised by the reviewer is an interesting topic for future studies.

3. The data presented in figure 3 are not entirely convincing due to the reliance on the observation of high-molecular weight bands as a readout of aggregation of insoluble DPRs. This data should be validated by performing fractionation of the soluble and insoluble fractions and quantification using meso scale discovery ELISA assays as described in the co-authors previous work (<https://doi.org/10.1186/s40478-020-01036-y>). Immunohistochemical analysis of fly brains should also be used to examine the number of DPR aggregates.

As suggested, we quantified soluble and insoluble GA levels by meso scale discovery (MSD) immunoassay in heads of untagged and tagged GA100 flies after 5 days of pan-neuronal, transgene expression. Surprisingly, we found no differences in soluble or insoluble GA levels between untagged and tagged GA100 flies (**Supplementary Fig 6C, D**), suggesting that the solubility of GA100 may not be changed by the protein tags. These findings are in contrast to our previous western blot results, which indicated increased levels of fluorescently tagged GA100, especially for the low molecular weight (LMW) bands (**Supplementary Fig 6A, B**), suggesting increased solubility of mCherry and GFP-tagged GA100. It is currently not clear why these two techniques led to different outcomes, especially since the same GA antibody was used for both assays. It is possible that despite the slow centrifugation speed used to prepare the western blot extracts, some of the most aggregated untagged GA100 may have been pelleted and therefore not been quantitatively loaded onto the gel. However, given that the two methods did not give

consistent results, we removed the conclusion about the increased solubility of GA100 upon tagging with mCherry/GFP from the manuscript. Nevertheless, we think it is worth to show the western blot results, as they are helpful to visualize the LMW degradation products, which cannot be addressed by the MSD immunoassay.

Finally, our previous pulse-and-chase western blot results indicated reduced stability of mCherry and GFP-tagged GA100. To further confirm these findings, we performed a new pulse-and-chase experiment using immunohistochemistry on fly brains expressing GA100 or GA100-mCherry. In agreement with our western blot results, the amount of GA signal in GA100 fly brains was unchanged following the 4-day chase (**Fig 4D-E**). However, while the detectability of the GA signal was lower in the brains of GA100-mCherry flies than in those of GA100 flies, the amount of GA signal in the former was significantly reduced during the chase (**Fig 4D-E**), suggesting that GA100-mCherry is less stable than untagged GA100. In addition, the mCherry signal in the brains of GA100-mCherry flies was reduced during the 4-day chase (**Fig 4D, F**). These results confirm that untagged GA100 is more stable than GA100-mCherry, which is likely to contribute to the degradation products that we detected by western blot and immunostainings.

Taken together, these results suggest that GA quantification is not trivial, as different assays did not point to the same results. While it is currently unclear whether the tags affect the solubility of GA levels, our results show that fluorescent tags lower the stability of GA, which may underlie its lower toxicity. The new data have been added to the manuscript (**Fig 4, Supplementary Fig S5 and Supplementary Fig S6**) and the findings are also included in the discussion.

4. Following on from the previous point, the authors should examine the effect of the tags upon localisation and morphology of the DPR "aggregates".

We have now used immunohistochemistry on fly brains to address this point. As outlined in more detail in our reply to point 1 of reviewer 1, we show that the mCherry tag does not affect the subcellular localization or morphology of PR and GR aggregates (**Fig 2A-B & Supplementary Fig S3A-F**). Furthermore, we show that the FLAG-tag did not affect the subcellular localization or aggregate appearance of GA100 (**Supplementary Fig S5A, B**). For GA100-GFP and GA100-mCherry, we only observed "aggregates" when using the fluorescent signal from GFP and mCherry, respectively (**Supplementary Fig S5A, B**). In contrast, anti-GA only revealed a diffuse cytoplasmic signal that did not overlap with the GFP/mCherry-positive aggregates (**Supplementary Fig S5A, B**). It has been reported previously in cell culture (Liu et al. 2022) that the GA antibody is unable to detect GA epitopes when they are fused to fluorescent tags, which probably explains why we did not see an overlap between the GA and GFP/mCherry signals. Noteworthy, we did not detect GA signal in the nucleus, independent of the presence or absence of a tag, suggesting that GA exclusively localizes to the cytoplasm in fly brains, consistent with our previous findings (Mizielinska et al. 2017; Morón-Oset et al. 2019). In summary, these results suggest that protein tags have no major influence on morphology and subcellular localization of the DPR aggregates.

5. None of the eye data (Fig1A-B, sup fig 1A-B) is quantified. One could argue that in both figures the GR100-mcherry eye looks less severely impacted than the untagged GR100, suggesting a very mild rescue. All eye data needs to be quantified using an appropriate scoring system. Given the multifaceted nature of these phenotypes a multi-point scoring system such as described by Pandey et al (2007) (<https://doi.org/10.1038/nature05853>) should be used.

As suggested, we quantified the eye phenotype upon expression of tagged and untagged DPRs using an appropriate scoring system in a blinded manner. The eye size of flies that expressed untagged or mCherry-tagged GA36 or GA100 did not differ from that of control flies either at 25°C nor at 29°C (**Fig 1C & Supplementary Fig S1C**). GR36-mCherry and PR36-mCherry flies had significantly smaller eyes than their untagged counterparts at 25°C and 29°C (**Fig 1C & Supplementary Fig S1C**), consistent with the data on larval survival (**Fig 1D & Supplementary Fig S1D**). In addition, we also found a statistically significant decrease in the eye size of PR100-mCherry flies compared to their untagged counterparts at 29°C (**Supplementary Fig S1C**), which is in line with our egg-to-adult viability results (**Supplementary Fig S1D**). In summary, our results of the eye phenotype quantification correspond well to the egg-to-adult viability measurements and support our previous conclusions on the effects of mCherry-tagging on the toxicity of the tagged GR and PR proteins. The eye size quantifications have now been included as **Fig 1C** and **Supplementary Fig S1C**.

Minor concerns

1. Control data in egg-to-adult viability assays is repeated between graphs. E.g. the data shown for WT and mCherry controls in figure 1C is identical to figure 1D. Data should not be re-used for independent experiments. If this is one experiment it should be graphed together. This is the same for the supplementary data.

We thank the reviewer for pointing this out. As suggested, we have now plotted the results on egg-to-adult viability into one common plot that comprises all tested genotypes assayed at 25°C (**Fig 1D**) and at 29°C (**Supplementary Fig S1D**) and added to the manuscript (**Fig 1D** and **Supplementary Fig S1D**).

2. Western blots should be repeated with anti-GFP/anti-mcherry/anti-FLAG to determine whether the tags remain attached or have been cleaved from the DPR. Cleavage of tags, especially GFP and mCherry has been commonly observed.

As suggested, we repeated the corresponding western blot experiments using anti-FLAG, anti-GFP and anti-mCherry antibodies (**Supplementary Fig S6A,B**). After short-term induction for 1 day (**Supplementary Fig S6A**), we only observed a single, low molecular weight (LMW) bands of the expected size for GFP-, mCherry and FLAG-tagged GA100. In addition to the LMW, we also observed a HMW band only for GFP- and mCherry-tagged GA100. After prolonged expression for 5 and 25 days, we were able to detect shorter bands in both GFP- and mCherry-tagged GA100 (**Supplementary Fig S6B**), suggesting partial cleavage of these fusion proteins, as has been previously observed (K. Zhang et al. 2021). The fact that the cleavage products were not recognized by the GA specific antibody, suggests that these are mainly derivatives of the GFP or mCherry protein. The western blots have been added to the manuscript as new **Supplementary Fig S6**.

3. The landing site for the new lines generated in this study needs to be included in the methods.

We have now included the information about the landing site in the material and method section and the corresponding figure legends.

Reviewer #3 (Comments to the Authors (Required)):

The toxicity of RAN translation products of the C9ORF72 expanded repeat is typically demonstrated using exogenously expressed proteins of a selected reading frame and with addition of a fluorescent protein or small epitope tag. Morón-Oset et al. use *Drosophila* to show that addition of either a fluorescent protein or an epitope tag can impact the toxicity of the RAN protein product. The authors first used expression of GA, GR or PR with 36 or 100 repeats and with or without a C-terminal mCherry tag in fly eye. They also used higher and lower expression levels. GA showed no effects between tagged and untagged but GR and PR dipeptide repeats (DPRs) disrupted eye morphology and reduced egg to adult survival and, importantly, the effects were stronger with a C-terminal mCherry tag. The results show that toxicity with regard to survival can be maxed out with 100 repeats and the impact of mCherry is not observed but mCherry particularly reduces survival upon expression of PR DPRs. A test of expressed DPR in adult neurons demonstrated that a C-terminal mCherry rescued the decreased life span observed with 36 or 100 GA DPRs. Overall the mCherry tag increased toxicity of arginine DPRs and reduced toxicity of GA DPRs in vivo. Their results also show that different tags have different effects on in vivo toxicity. GFP and mCherry completely blocked lifespan shortening of GA DPRs and the short Flag tag reduced but did not eliminate lifespan shortening. Overall there was a clear modifier effect of a fluorescent or small epitope tag. Importantly the authors show that differences on in vivo toxicity does not correlate with levels of protein expression but do correlate with propensity to aggregate into high molecular weight species on a denaturing gel. Using p62 upregulation as a measure of cell stress following induced expression of different DPR forms in neurons, the authors show that large fluorescent protein tags but not a FLAG tag, prevented p62 upregulation by GA DPRs. Similarly a TUNEL assay showed that GA DPRs with and without Flag increased DNA fragmentation but an mCherry tag prevented this effect.

The main finding that tags on DPRs impacts toxicity is an important point and the authors present evidence that aggregation, p62 upregulation for GA DPRs and toxicity due to DNA damage is involved. The role of aggregation is strongly suggestive but not definitive since it is based on a signal on western blots what appears to be just below the bottom of the well. One concern is that the protein samples were centrifuged prior to loading the supernatant so that aggregated proteins could be lost prior to the western and comparisons only on the westerns might not reflect differences in aggregation.

We thank the reviewer for this consideration. While we cannot fully exclude that some aggregated protein was lost due to the centrifugation, the RIPA fraction was pipetted up and down several times before loading protein samples onto protein gels in order to minimize this effect. This information has now been included in the material and method section.

Another somewhat minor concern is the transition through the manuscript of testing multiple DPRs to only GA DPRs. The results address the potential basis for discrepancies regarding GA DPR toxicity, which is important. There are different effects of the same tags

on different DPRs that are not addressed mechanistically. The mechanisms of these effects are of interest but not critical.

We agree with the reviewer that it is potentially interesting to address how the same protein tag can cause such different effects in GA and the arginine-rich DPRs, i.e., decreased vs. increased toxicity, respectively. However, the main finding of this study is that commonly used protein tags can affect the toxicity of DPRs, which is important when designing new animal models to study DPR function. To highlight this point, an understanding of the underlying mechanisms whereby protein tags affect toxicity is not absolutely essential. Nevertheless, we conducted further experiments based on the comments by reviewers 1 and 2, to gain further insights about how protein tags might affect DPR toxicity. By measuring protein levels of the arginine-rich DPRs, we showed that while the mCherry-tag did not affect the protein level of GR (**Fig 2C, D**), PR levels were increased upon mCherry-tagging (**Fig 2E**). Thus, the same tag can even cause different effects on arginine-rich DPRs and the increased toxicity of PR might be explained by its higher protein levels. We also conducted further experiments concerning the loss of toxicity upon mCherry-tagging of GA and show that GA100-mCherry is preferentially degraded by the proteasome (**Supplementary Fig S8**). In summary, the effects of tags on toxicity are DPR-specific and probably affect different underlying mechanisms. This conclusion has been added to the discussion.

May 26, 2023

RE: Life Science Alliance Manuscript #LSA-2022-01739R

Prof. Linda Partridge
Max Planck Institute for Biology of Ageing
Biological Mechanisms of Ageing
Joseph-Stelzmann Str, 9b
Cologne 50931
Germany

Dear Dr. Partridge,

Thank you for submitting your revised manuscript entitled "Toxicity of C9orf72-associated dipeptide repeat peptides is modified by commonly used protein tags". We would be happy to publish your paper in Life Science Alliance pending final revisions necessary to meet our formatting guidelines.

- please address Reviewer 2's remaining comments
- please rename: Legends of main figures" section with "Figure legends" and "Legends of supplementary legends" with "Supplementary figure legends"
- please use the [10 author names, et al.] format in your references (i.e. limit the author names to the first 10)
- please add ORCID ID for corresponding (and secondary corresponding) author--you should have received instructions on how to do so
- please add the Twitter handle of your host institute/organization as well as your own or/and one of the authors in our system
- please add a callout for Figure S2A,B to your main manuscript text

A. FINAL FILES:

B. MANUSCRIPT ORGANIZATION AND FORMATTING:

Thank you for your attention to these final processing requirements. Please revise and format the manuscript and upload materials within 5 days.

Sincerely,

Reviewer #1 (Comments to the Authors (Required)):

The authors have fully address the previous comments and concerns.

Reviewer #2 (Comments to the Authors (Required)):

I commend the authors on addressing the vast majority of the issues raised by the reviewers. I believe however it is important, reflecting on their results and their own conclusions, that they scale back the claims made in the abstract that "Tagging of arginine-rich DPRs with mCherry increased toxicity". Their data does not show an effect on 100 repeat arginine constructs suggesting that the effect in arginine models may be specific to shorter repeat models. In figure 1 there is only a significant increase in toxicity in the 36 repeat lines, not the 100 repeats for viability, eye toxicity and longevity. This in itself is an important finding. In order to avoid misinterpretation of their paper, particularly by the majority of people who only skim the abstract I ask the authors to alter their abstract to state that Tagging of 36 arginine-rich DPRs with mCherry increased toxicity, whilst tagging of 100 repeats did not.

Minor

At line 370-372 the authors state that "Expression of GA175-GFP in immortalized cells (May et al. 2014), GA50-FLAG in yeast (Jovičić et al. 2015; Boeynaems et al. 2016), GA1000-GFP in Drosophila (West et al. 2020) and GFP-polyGA in primary cortical and motor neurons (Wen et al. 2014) caused no toxicity"

- For May et al 2014 I do not see where they found this data. From the study I where they looked at toxicity in GA175-GFP expressing cells. I can only find where they did apoptosis and shall analysis in primary neurons expressing my tagged GA-149. I may have just missed the data but please check the statement is correct.

- For West et al 2020 the GA1000 flies do show toxicity as observed in figures 8 and 10 looking at climbing ability.

June 3, 2023

RE: Life Science Alliance Manuscript #LSA-2022-01739RR

Prof. Dame Linda Partridge
Max Planck Institute for Biology of Ageing
Biological Mechanisms of Ageing
Joseph-Stelzmann Str 9b
Cologne, NRW 50931
Germany

Dear Dr. Partridge,

Thank you for submitting your Research Article entitled "Toxicity of C9orf72-associated dipeptide repeat peptides is modified by commonly used protein tags". It is a pleasure to let you know that your manuscript is now accepted for publication in Life Science Alliance. Congratulations on this interesting work.

DISTRIBUTION OF MATERIALS:

Again, congratulations on a very nice paper. I hope you found the review process to be constructive and are pleased with how the manuscript was handled editorially. We look forward to future exciting submissions from your lab.

Sincerely,
